



# New flood frequency estimates for the largest river in Norway based
# on the combination of short and long time series
Kolbjørn Engeland[1], Anna Aano[1,2], Ida Steffensen[3], Eivind Støren[3,4], Øyvind Paasche[4,5]
[1]The Norwegian Water Resources and Energy Directorate, Oslo, Norway
[2]UDepartment of Geosciences, University of Oslo, Norway
[3]Department of Earth Science, University of Bergen, Norway
[4]Bjerknes Centre for Climate Research, Bergen, Norway
[5]NORCE Climate, Bergen, Norway
*Correspondence to*: Kolbjørn Engeland (koe@nve.no)
**Abstract.** The Glomma river is the largest in Norway with a catchment area of 154 450 km$^2$. People living near the shores of
this river are frequently exposed to destructive floods that impair local cities and communities. Unfortunately, design flood
predictions are hampered by uncertainty since the standard flood records are much shorter than the requested return period and
also the climate is expected to change in the coming decades. Here we combine systematic- historical and paleo-information
in an effort to improve flood frequency analysis and better understand potential linkages to both climate and non-climatic
forcing. Specifically, we (i) compile historical flood data from the existing literature, (ii) produce high resolution X-ray
fluorescence (XRF), Magnetic Susceptibility (MS) and Computed Tomography (CT) scanning data from a sediment core
covering the last 10 300 years, and (iii) integrate these data sets in order to better estimate design floods and assess non-
stationarities. Based on observations from Lake Flyginnsjøen, receiving sediments from Glomma only when it reaches a certain
threshold, we can estimate flood frequency in a moving window of 50 years across millennia revealing that past flood
frequency is non-stationary on different time scales. We observe that periods with increased flood activity (4000-2000 years
ago and <1000 years ago) corresponds broadly to intervals with lower than average summer temperatures and glacier growth
whereas intervals with higher than average summer temperatures and receding glaciers overlap with periods of reduced number
of floods (10 000 to 4000 years ago and 2200 to 1000 years ago). The flood frequency shows significant non-stationarities
within periods having increased flood activity as was the case for the 18th century, including the AD 1789 ('Stor-Ofsen') flood
being the largest on record for the last 10 300 years at this site. Using the identified non-stationarities in the paleoflood record
allowed us to estimate non-stationary design floods. In particular, we found that the design flood was 23% higher during the
18$^{th}$ century than today and that long-term trends in flood variability are intrinsically linked to the availability of snow in late
spring linking climate change to adjustments in flood frequency.
Keywords: flood, lakes, extremes, paleofloods, Norway, non-stationarity.





**1 Introduction**
Floods are among the most widespread natural hazards on Earth. The impacts, destruction and costs associated with hazardous
floods are increasing in concert with climate change, a trend that are most likely to strengthen in the decades to come (e.g.
Alfieri et al., 2017; Hirabayashi et al., 2013; IPCC, 2012) In Europe, spatial flood patterns are changing both in terms of timing
and magnitude (Blöschl et al., 2017, 2019) challenging us to examine new ways to interlink not only different types of data,
but also flood information on different time scales. Earlier studies have shown that uncertainties can be reduced if, for instance,
historical data are included in estimation of floods with long return periods (e.g. Brázdil et al., 2006a; Engeland et al., 2018;
Macdonald et al., 2014; Payrastre et al., 2011; Schendel and Thongwichian, 2017; Stedinger and Cohn, 1986; Viglione et al.,
2013). Here we seek to extend the possibility of using historical data by including time series of reconstructed floods based on
lake sediment archives which can retain imprint of past flood activity (Gilli et al., 2013; Schillereff et al., 2014; Wilhelm et
al., 2018). The ultimate goals of this exercise are to (i) reduce uncertainty associated with flood prediction and (ii) provide
additional insight to flood variability on longer time scales, and thereby improve our understanding of how climate change
impacts floods.
In many European countries, flood mitigation measures aim to reduce the exposure and vulnerability of the society to
floods. Examples of such measures can include reservoirs, flood safe infrastructure, and land-use planning in flood-exposed
areas. These mitigation measures require estimates of design floods, i.e. the flood size (typically given in $m^3/s$ for a specified
annual exceedance probability (AEP) or return period (RP). The required design AEP or RP depends on the impact of a flood.
The Norwegian building regulations (TEK17, 2018) exemplifies this. They require that that buildings of particular societal
value such as hospitals should be able to resist or be protected from at least a 1,000-year flood whereas normal settlements
should withstand 200-year flood and storage facilities at least a 20-year flood. Design flood estimates are commonly based on
analysis of the frequency and magnitudes of observed floods using measurements derived from a streamflow gauging station.
Recall that for many applications, estimates of 200- up to1000-year floods are required (see Lovdata (2010) and TEK17
(2018)for regulations in Norway). This is not a trivial task for at least two reasons. Firstly, we have limited amount of data and
the estimation uncertainty for a 1000-year flood is large with only 50-100 years of data. Secondly, we plan for the future (i.e.
for the life time of a construction), but in many cases it can be necessary to account for non-stationarities in floods caused by
past as well as anticipated future changes in climate.
Both challenges can be addressed by using data covering longer time periods including historical data (e.g. Benson,
1950; Brázdil et al., 2006b; Macdonald et al., 2014; Schendel and Thongwichian, 2017; Viglione et al., 2013) and/or paleoflood
data (e.g. Benito and O'Connor, 2013). The fact that sediment deposits can be unambiguous evidence of past floods is
documented in many studies since 1880 AD (Bretz, 1929; Dana, 1882; Tarr, 1892), and an early example of how to estimate
discharge associated with giant paleofloods can be found in Baker (1973) whereas paleoflood hydrology as a concept and
terminology was first introduced by Kochel and Baker (1982).
In order to include information about past floods in flood frequency analysis, it is necessary to estimate the flood sizes
in $m^3/s$. A successful approach for assessing the stage and the volumes for paleofloods is to use slack-water deposits along
river canyons (e.g. Baker, 1987, 2008; Benito and O'Connor, 2013; Benito and Thorndycraft, 2005). Following this approach,
water level during floods can be deduced from the elevation of the deposits enabling hydraulic models to estimate flood
volumes for specific events. During the recent 20 years, lacustrine sediments has proven to be another reliable source of
paleofloods (Gilli et al., 2013; Schillereff et al., 2014; Wilhelm et al., 2018). Sediment cores retrieved from lakes that
periodically receive sediments delivered by floods can be used to extend local hydrological time series spanning thousands of
years. Since lake sediment archives for the most are continuous records, they can complete the snapshot information provided
by flood terraces still present in the landscape or anecdotal information about historical floods.
Lakes fit for using lacustrine sediments to analyze flood frequencies are typically found where (i) flood sediments are
preserved at the bottom of lakes (ii) there is a detectable on/off signal for sediments left by floods, and (iii) a distinct contrast





between flood deposits and regular background sedimentation (Gilli et al., 2013). Detection of flood layers in the cores can be
based on X-ray fluorescence (XRF) scanning (e.g. Czymzik et al., 2013; Støren et al., 2016) magnetic susceptibility (MS)
measurements (e.g. Støren et al., 2010), computed tomography (CT) scanning (e.g. Støren et al., 2010), or spectral reflectance
and color imaging (Debret et al., 2010).

5       There are multiple sources for historical flood data including (i) annals, chronicles, memory books and memoirs; (ii)

weather diaries; (iii) correspondence (letters); (iv) special prints; (v) official economic and administrative records; (vi)
newspapers and journals; (vii) sources of a religious nature; (viii) chronogramme; (ix) early scientific papers, compilations
and communications; (x) stall-keepers' and market songs; (xi) pictorial documentation; and (xii) epigraphic sources (Brázdil
et al., 2012). Depositories of historical flood data are listed in Brázdil et al. (2006a) and Kjeldsen et al. (2014). An overview
over historical floods in Norway is available in Roald (2013). For quantitative analyses, it is nonetheless necessary to find
evidence of historical flood stages and estimate flood discharge based on hydraulic calculations (Benito et al., 2015).
The paleo- and historical flood information can be used – in combination with systematic data – to estimate design
floods (see e.g. Engeland et al., 2018; Kjeldsen et al., 2014; Stedinger and Cohn, 1986). To include the paleo- and historical
information in flood frequency analysis, we also need to know all floods exceeding a fixed threshold during a specified time
interval. Several studies demonstrate that, given that the fixed threshold is high enough, it is adequate to know the number of
floods exceeding this threshold in order to improve flood quantile estimates (Engeland et al., 2018; Martins and Stedinger,
2001; Payrastre et al., 2011; Stedinger and Cohn, 1986). A Bayesian approach to flood frequency analysis with historical- and
paleodata sources was introduced by Stedinger and Cohn (1986) and Gaál et al. (2010). This approach allows, in a flexible
way, the introduction of multiple fixed thresholds and data sources, and is therefore well suited for combining systematic-
historical and paleo- data in a joint flood frequency analysis.
When we predict flood frequency for the future, the standard assumption is stationarity or put differently: it's assumed
that the period with instrumental data is representative for the future.  In many cases, when the analysis is based on flood data
from a streamflow gauging station covering a limited period, it is a robust assumption (Serinaldi and Kilsby, 2015).  However,
in the face of expected changes in climate, it is useful to take into account the risk for floods in the future (Hanssen-Bauer, I.
Førland, E. J. Haddeland et al., 2017; Lawrence, 2020; Paasche and Støren, 2014). For Norway, tailored guidelines for adaption
to future flood risk  are provided by the Norwegian Center for Climate Services (https://klimaservicesenter.no/) based on
results from climate projection studies (Lawrence, 2020).  A current practice is to use flood inundation maps where estimated
future flood levels for specific return periods are shown (e.g. NVE flood zone maps, 2020;  Orvedal and Peereboom, 2014).
Such maps are commonly used in land-use planning.
Since the historical- and paleodata covers much longer time periods than streamflow data, they can be an excellent
source for non-stationarity in actual flood sizes and the underlying flood generating processes. One approach is to link the
frequency of floods to the underlying climatic drivers (e.g. mean temperature, precipitation and large-scale circulation patterns
(e.g. Gilli et al., 2013; Kjeldsen et al., 2014; Støren et al., 2012; Støren and Paasche, 2014). A major challenge when using
paleo- and historical flood information, is precisely to disentangle non-stationarity in climatic drivers from non-stationarities
caused by changes in land-use and/or the 'archiving processes' of the data. Changes in land-use can, for instance, be related to
farming practices and timber-logging. Changes in the archiving process might be caused by changes in the perception threshold
that depend on societal development (Kjeldsen et al., 2014; Macdonald and Sangster, 2017). Also changes in the river channel
might limit the possibility to estimate the magnitude of paleo- and historical floods (Brázdil et al., 2011).
The primary objective of this paper is to combine systematic- historical and paleo-information in a flood frequency
analysis in order to better understand and predict changes in flood frequency and magnitude for Norway's largest river,
Glomma. In particular we want to explore:
• Past variability in floods as reconstructed from lake sediment cores.
• Potential non-stationarity in our new paleoflood record and its potential connection to regional climate change.





• The added value of combining systematic-, historical-, and paleo-flood data when estimating flood quantiles.
• Potential non-stationarities in design floods.
The unique contribution of this study is thus to combine three different information sources in an attempt to improve flood
frequency estimations and better understand the underlying mechanisms that cause significant changes in flood variability over
time.
**2 Study area**
**2.1 Study catchment**
The target site for this study is the city Elverum lying next to the river Glomma having an upstream catchment area of 154 450
km$^2$ (Fig. 1). The elevation in the catchment ranges from 180 masl at Elverum to 2178 masl at Mt. Rondslottet in Rondane
further north, and is covered by forest (52 %), open areas above the timber line (27%), bogs (10 %), lakes (3%), and agricultural
areas (2%). Only 0.13% is represented by urban areas. The average annual precipitation is 580 mm with the summer months
being the wettest. The annual average temperature is -0.65 ℃ but the climate is continental. January has the coldest month
with -11.2 ℃ whereas July is the warmest with 10 ℃. The low winter temperatures result in a considerable seasonal snow
cover which has a direct impact on the streamflow. Minimum flows are observed during winter (December – April) whereas
the highest flows take place during the snow melt season (May – June), as shown in Fig. 2. The main flood season occurs
during the snow melt season (May – June) with the rare exception of a few minor floods that arrive during the autumn season
due to long duration intense rainfall.

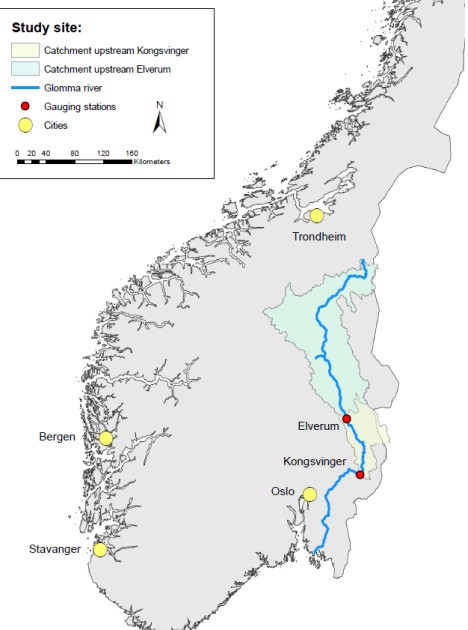

Figure 1: The location of the streamflow gauging station at Elverum used for flood frequency analysis, and the site for paleodata
collection close to Kongsvinger.
The catchment has several hydropower reservoirs with a total regulation capacity presently around 10% of the average
annual runoff. The first reservoir was built in 1913, and since 1937 this and other reservoirs have resulted in decreased flood





sizes (Pettersson, 2000). The monthly flows during winter has increased and most flood peaks have decreased after 1937 (Fig.
2). The catchment has undergone noteworthy land-use changes during the last 400 years. In the 17-19th century, the forest
areas were reduced due to mining, timber export and farming practices. Since the beginning of the 20th century, the forest
covered areas have increased slightly whereas the timber volume has increased substantially mainly due to farming and forestry
practices e.g. reduced grassing of domestic livestock and forestation, (Grønlund et al., 1999).

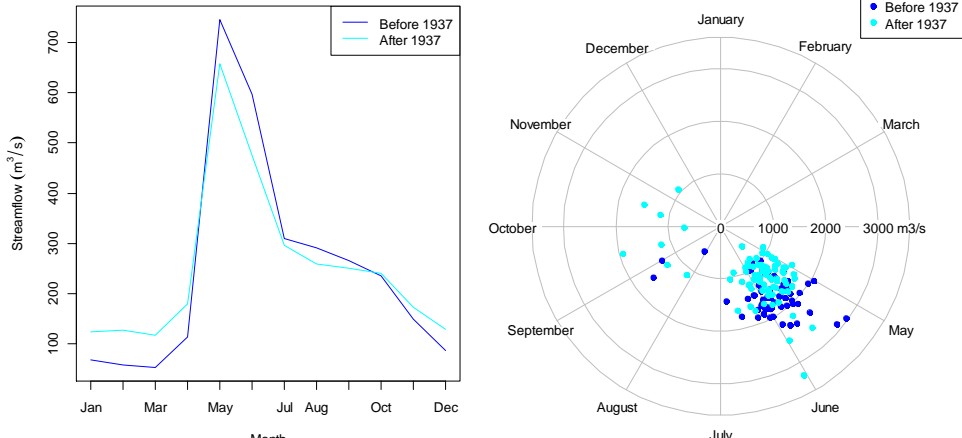

Figure 2: Seasonality of Glomma's monthly streamflow (left) and annual maximum floods (right) at Elverum. The dampening
of floods after 1937 is explained by up-stream dam-building.
**2.2 Study site for paleodata**
To establish a flood record covering most of the Holocene (<11 700 years), two sediment cores were retrieved at 16 m water
depth from Flyginnsjøen (UTM: 33V 0337459 6670202) located close to Kongsvinger, around 80 km south of Elverum as the
crow flies (Fig. 1). A detailed map of the study area is shown in Fig.3 and conceptual model of the lakes involved, flood water
levels, thresholds and flood pathways are shown in Fig. 4. During normal conditions, water flows from Tavern and Vingersjøen
(catchment area 72.0 km$^2$) into Glomma. When the streamflow in Glomma exceeds 1500 m$^3$/s, the flow direction reverse, and
around 1-2 % of the water flows from Glomma and over to Vingersjøen and further into Tavern, Flyginnsjøen, leaves the
Glomma catchments and follows the river Vrangselva across the border to Sweden (Pettersson, 2001). These bifurcation events
enable flood water from Glomma to reach Flyginnsjøen where part of the suspended sediment load is deposited. This is in
stark contrast to 'normal conditions' for the lake, when the minerogenic sediment delivery is marginal compared to the organic
material, as outlined below. The repeated increase in discharge during floods, remobilize readily available sediments –
originating mainly from the last deglaciation – and allow for the subsequent deposition of fine-grained minerogenic material.
Bathymetric map of Lake Flyginnsjøen and the coring sites which were chosen at the deepest part of the lake, close to the inlet
is shown in Fg. 5. For addition details about the study site, and its surroundings, see the master theses by Aano (2017), Follestad
(2014), and Steffensen (2014).





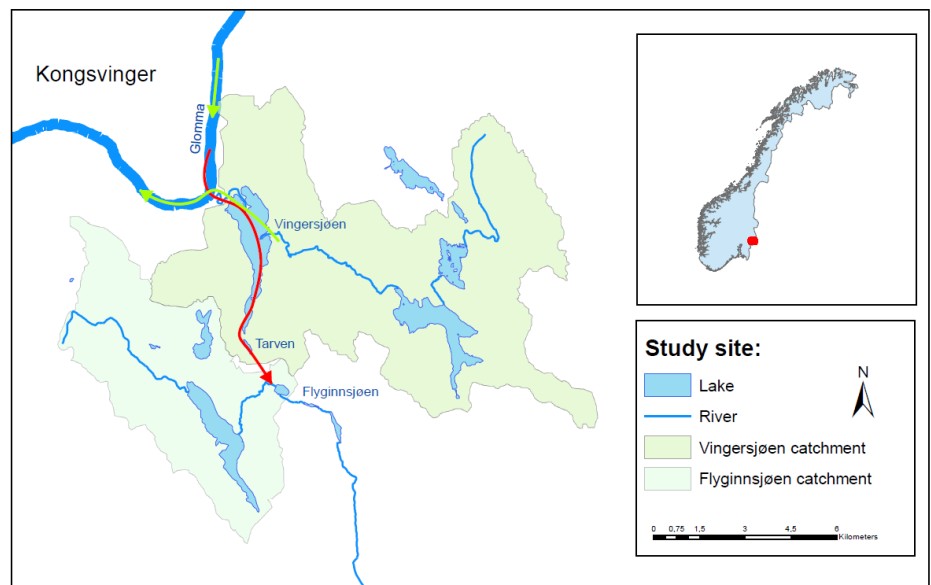

Figure 3: Study site for the paleodata. The sediment cores were extracted from lake Flyginnsjøen. The green arrows indicate
the flow direction under normal conditions, whereas the red arrow shows the flow direction whenever there is a flood that
exceeds 1500 m$^3$/s and bifurcation occurs.

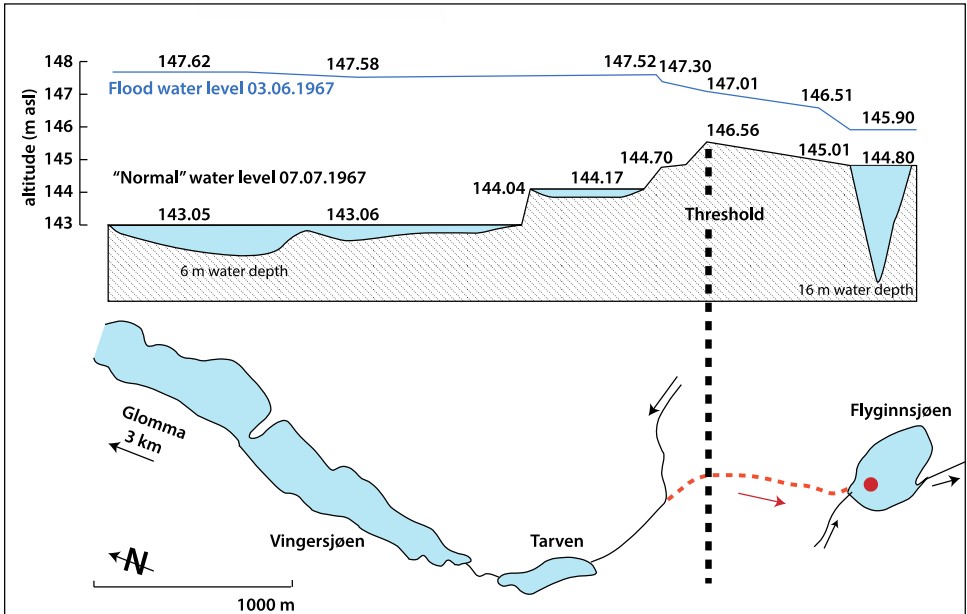

Figure 4: Schematic model of the lakes involved, flood water levels, thresholds and flood pathways (After Hegge, 1968).
The example shows observed water level exceeding the threshold during the flood in 1967 (2533 m$^3$/s), and the normal water
level approx. one month after the flood event.  The dotted red line and arrow show the bifurcation over the threshold, and the
red point marks the coring site in Flyginnsjøen.





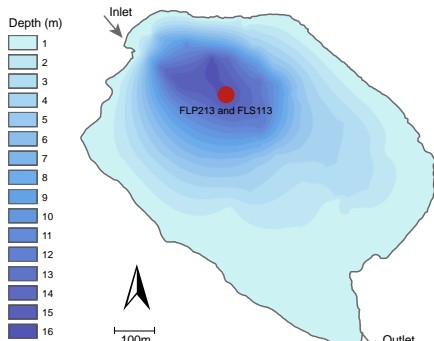

Figure 5. Bathymetric map of Lake Flyginnsjøen and the coring sites which were chosen at the deepest part of the lake, close
to the inlet.
**3 Data sources and Methodology**
**3.1 Systematic flood data**
Annual maximum flood at Elverum (station number 2.604) for the period 1871-1937 was used for the flood frequency analysis.
For this period, we assumed that the flood data were not significantly affected by river regulations (Pettersson, 2000). The
modern observations are shown in Fig. 6 together with the known historical floods as well as annual maxima daily floods from
the period after 1937, when we observe a minor decrease in average flood size after 1937.

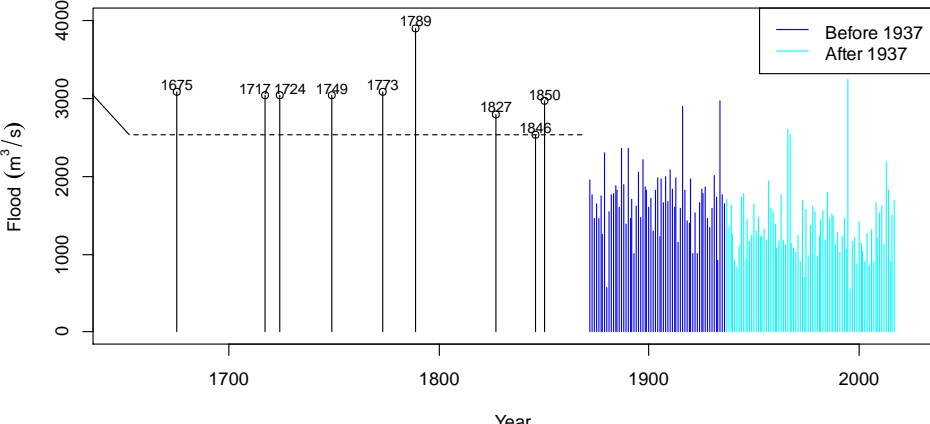

Figure 6: Systematic and historical flood data at Elverum. The 1789 AD flood known as *Stor-Ofsen* in Norway stand out in
this record.
**3.2 Historical flood data**
Historical flood information back to 1675 is available as water levels marked at a flood stone close to Klokkerfossen ('fossen'
meaning waterfall) at the Norwegian Forest Museum in Elverum (Fig. 7 and Table 1-2). Table 1 lists the water levels and
discharges for floods exceeding the 1967 flood are highlighted on the flood stone which was erected in 1968. The water levels
were carved into the stone in 1969 based on recommendations from NVE (Hegge, 1969); the 1995 flood was added later.





There is another flood stone nearby at Grindalen (also shown in Fig. 7). It was erected as early as in 1792 in order to remember
the floods of 1773 and 1789, which were large indeed.
The flood stone at Grindalen is 2 km upstream the flood stone at Klokkerfossen with the streamflow gauging station
at Elverum in the middle. A waterfall at Klokkerfossen is the controlling profile for the water levels at all three locations.
Hegge (1969) developed relationships between water levels at the Elverum gauging station and the flood stone at Prestfossen
shown here in Table 1. The water levels at Elverum gauging stations were transformed to discharges by using the local rating
curve. In this study, we included all floods exceeding the observed 1967 flood peak at 2533 m$^3$/s in the flood frequency
analysis. By following this approach, we are confident that we only include information about all floods exceeding a specific
flood level.
Table 2 summarizes the available historic information and important sources for these floods. The floods in 1675,
1717, and 1749 are all described in Finne-Grønn (1921) and Otnes (1982) whereas information for the flood mark in 1724 is
not found in any written source. Detailed information about water levels for floods prior to 1773 were estimated in the absence
of historical data. The water levels in 1773, 1789, 1827 and 1846 are all engraved in the flood stone in Grinsdalen and employed
here as a basis for calculating the water level at the Elverum gauging station and also for the flood stone at Klokkarfossen.
Having said that, we still include all flood water levels listed in Hegge (1969). More information about historical flood of the
Glomma River and at Elverum is provided by Finne-Grønn (1921), Otnes (1982), and Roald (2013). During the period 1675
to 1870, we see that 8 floods exceeded the observed 1967 flood peak at 2533 m$^3$/s. The 18th century has a large number of
floods at this location. All floods occurred in late May with the notable exception of *Stor-Ofsen* in 1789 which occurred in late
July.
The largest historical flood in this region was *Stor-Ofsen* which took place in 22-23 July 1789 when peak discharge
reached 3900 m$^3$/s at Elverum (GLB, 1947) being only slightly smaller than our estimate (see Table 1). Numerous catchments
in eastern Norway flooded at the time resulting in 61 fatalities, destruction of infrastructure, farms and crops. The economic
losses were extraordinary and in the aftermath of the flood, around 1500 farms got tax reduction (Otnes, 1982).
Table 1: Water levels at Elverum gauging station and at the flood monument from Hegge (1969). The various streamflow
peaks are constructed based on the rating curve at the gauging station 2.119 and rating curve period 1881-1970.

| Date | Height – gauging station (m) | Height – flood monument (m) | Streamflow (m$^3$/s) Peaks |
|---|---|---|---|
| 28.05.1675 | 4.50 | 3.35 | 3141 |
| 24.05.1717 | 4.30 | 3.22 | 2963 |
| 17.21.1724 | 4.25 | 3.19 | 2919 |
| 24.05.1749 | 4.20 | 3.16 | 2875 |
| 30.05.1773 | 4.55 | 3.38 | 3187 |
| 22.07.1789 | 5.35 | 3.86 | 3944 |
| 27.05.1827 | 4.04 | 3.06 | 2736 |
| 24.05.1846 | 3.87 | 2.95 | 2592 |
| 25.05.1850 | 4.33 | 3.24 | 2989 |
| 11.05.1916 | 4.30 | 3.22 | *2892* |
| 08.05.1934 | 4.36 | 3.26 | *2963* |
| 20.05.1966 | 3.90 | 2.97 | *2600* |
| 02.06.1967 | 3.87 | 2.95 | *2533* |
| 02.06.1995 | --- | --- | *3238* |


Table 2 Information about large historical floods at Elverum.

| Date | Information | Source |
|------|-------------|--------|
| 28.05.1675 | Large flood in Elverum used as a reference for later floods. | Finne-Grønn (1921) Otnes (1982) |
| 24.05.1717 | The largest flood since 1675 | Finne-Grønn (1921) Otnes(1982) |
| 1724 | No information found | |
| 24.05.1749 | Large amounts of snow during winter. The flood was smaller than in 1675 and similar to the floods in 1717 and 1724. The flood peaked around 12:00. | Finne-Grønn (1921) Kvernmoen and Kvernmoen(1921) |
| 29-30.05.1773 | Highest flood in man's memory and higher than in 1675. The whole village flooded. Marked at flood stone in Grindalen | Finne-Grønn (1921) Kvernmoen and Kvernmoen(1921) |
| 22-24.07.1789 | The flood peaked between 22:00 and 24:00 the whole village at Elverum destroyed. Marked at flood stone in Grindalen. | GLB (1947) |
| 27.05.1827 | 2.5 alen (156 m) lower than 1789 and 0.5 alen (31.3 cm) lower than 1773. Almost the whole village was flooded. Marked at flood stone in Grindalen. | Otnes (1982) |
| 26.05.1846 | Marked at flood stone in Grindalen. | Roald (2013) |
| 24-26.05.1850 | Marked at flood stone in Grindalen. | Roald (2013) |

Figure 7: Map on the left shows the locations of the flood stones and the gauging station at Elverum (left). Pictures to the right
shows the flood monuments at Grindalen (middle, photo: N.R. Sælthun) and the Norwegian forest museum (right, photo: Ø.
Holmstad).





1         Prior to *Stor-Ofsen*, there was a substantial amount of snow in the mountains, deep soil frost, and rainfall that had

saturated the soil. During the actual flood event, warm and humid air masses from the southeast were blocked by colder air
masses in the north-west, resulting in high rainfall over the entire region. The rainfall intensity peaked on the 22 July. The
flood started on 21 July in small brooks and culminated the following day (Østmoe 1985). The main rivers at the bottom of
the valleys rose to unprecedented levels and the flood was also accompanied by numerous landslides. The water levels of this
flood are known from several markings cut into rocks, and many flood levels have later been transferred to monuments erected
at locations near the major rivers (Engeland et al., 2018; Finne-Grønn, 1921; Otnes, 1982; Roald, 2013).
*3.2.1 Bifurcation events*
Descriptions of bifurcation events and lists of estimated flow volumes in Glomma at Kongsvinger are found in Aano (2017),
Pettersson (2001), Hegge (1968), and Reusch (1903). From 1851 to 2013, 79 events in 77 different years were recorded. In
1957 and 1987 there were bifurcation events both in the spring and in the autumn; 4 of the 79 events occurred during the
autumn. For the interval between 1953-2013, the same period that is covered by FLS113, there were 22 bifurcation events.
The transferred volume for the period 1851-2013 is presented in Fig. 8. The five years with the largest transferred volumes are
1916, 1934 1966, 1967 and 1995 with corresponding peak floods at Elverum yielding 2892, 2963, 2600, 2533, and 3238 $m^3/s$,
respectively. Note that there is a strong statistical correlation (rsq=0.94) between transferred volume and the maximum
transferred discharge. In addition to actual discharge of the individual floods, the duration of each bifurcation event determines
the total volume. The estimated peak bifurcation discharge in 1995 was substantially smaller than the estimate for 1916, despite
the fact that the water level in Glomma was somewhat higher in 1995 (Pettersson, 2001). Possible explanations for this minor
discrepancy are that increased vegetation and/or a local road bridge has reduced the capacity of the intermittent water course.
The number of events has decreased since around 1930, mainly due to construction of hydropower reservoirs.

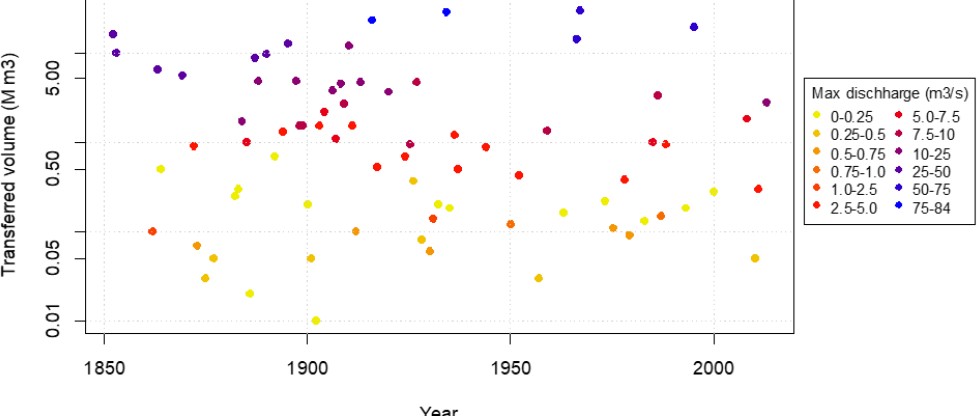

Figure 8: Transferred volume (M $m^3/s$) and maximum discharge ($m^3/s$) indicated by color for bifurcation events at
Kongsvinger. Estimates are obtained from Aano (2017), Pettersson (2001), Hegge (1968), and Reusch (1903).
**3.3 Paleohydrological flood data from lakes**
*3.3.1 Identification of sediment layers*
Two sediment cores were retrieved from Flyginnsjøen in 2013 (see Sect. 2.2). Coring sites shown in Fig. 5 were selected were
chosen at the deepest part of the lake based on a bathymetric survey of the lake using a Garmin Fishfinder echo sounder, and





sediment cores were retrieved using a 110 mm diameter piston corer (FLP213) (Nesje, 1992) and an HTH gravity corer
(FLS113) (Renberg and Hansson, 2008) Samples of 1 cm$^3$ were extracted at 0.5 cm intervals from the sediment cores, dried
overnight at 105 ºC, and weighed to measure dry-bulk density (DBD) (Blake and Hartge, 1986). The same samples were
subsequently burned at 550ºC to measure the weight loss-on-ignition (LOI) as an estimate of the organic matter content (Dean,
1974). Geochemical properties of the sediment cores were measured using a Cox Analytics ITRAX XRF core scanner at 200
µm resolution, running a Cr X-ray tube at 30 kV and 45mA for 10 sec. measurements at each step. XRF measurements were
normalized against total scatter (incoherent and coherent) to reduce potential influence of water content. Images of the split
core surface was also captured by the ITRAX core scanner, and 8-bit (255 values) Black & White (BW) values were obtained
from a 75 pixel wide average along the length of the core at 200 µm resolution using Image J software. A ProCon Alpha Core
Computed Tomography (CT) scanner running at 100 kV, 200 mA for 250 ms was used to generate 3D X-ray imagery of
FLS113 with a voxel resolution of 80 µm. CT data was reconstructed using ring-artefact and median filter in the Volex CT
Offline software (ProCon X-ray gmbh), and visualized in Avizo Fire 9.1 (FEI) software. The CT data is given as 16-bit (65636
values) grayscale values, interpreted to indicate relative densities due to minimal photoelectric effect at 100 kV (Wellington
and Vinegar, 1987) and extracted at 80 µm resolution through a centreline of the FLS113 sediment core. MS was measured
on the surface of the split sediment cores at 2 mm sample intervals with a Bartington MS2E point sensor using the CoreSusc
MkIII core scanner.

17       The area between Vingersjøen and Flyginnsjøen (Fig. 4) is rich in glaciofluvial deposits easily remobilized whenever

floods occur. Bifurcation events in Glomma causes precisely such a fundamental change in the erosion regime in this area,
causing river-flooding in a normally dry area (see Sect. 3.2.1). The following calculations and interpretations are thus based
on the assumption that bifurcations events can be recorded as marked increase in minerogenic input to lake Flyginnsjøen,
redeposited from the pre-existing glaciofluvial deposits in the catchment.

22       To quantify the frequency of such events a local peak detection algorithm was applied on parameters sensitive to

changes in minerogenic input. Flood deposits was defined as peaks in the measured parameters where (*i*) the measured
concentration is higher than the two surrounding values, (*ii*) the difference between the peak and the lowest value within a
specified time window (*w*) exceeds a specified threshold $h_1$ and, (*iii*) the difference between the peak and the lowest value at
each sides of the peak (within the time window) exceeds a specified threshold $h_2$ where $h_2 < h_1$. Each peak should be separated
by at least 9 months.

28       To produce a Holocene flood record based on the sediment cores from Flyginnsjøen depth in core was transformed

to time using Bacon age-depth modelling software (Blaauw and Christeny, 2011) (see Sect. 4.1.1), and frequency of events in
50-year moving window was quantified. In order test to what extent the lake sediment records reproduce modern and historical
observations, identified flood layers was compared to with instrumental streamflow data.

### 3.4 Flood frequency modelling

*3.4.1 Stationary flood frequency modelling*

A Generalized Extreme Value (GEV) distribution was invoked to establish a flood frequency model for floods at Elverum.
The GEV distribution is shown to be a limiting distribution for block maxima (Embrechts et al., 1997; Fisher and Tippett,
1928; Gnedenko, 1943):

$$F(x) = \begin{cases} exp\left\{-\left[1 - k\left(\frac{x-m}{\alpha}\right)\right]^{1/k}\right\} \\ exp\left\{-exp\left(-\frac{x-m}{\alpha}\right)\right\} \end{cases} \qquad (1)$$






Where $m$ is a location parameter, $\alpha$ a scale parameter and $k$ a shape parameter. We estimated the parameters using a Bayesian approach. Their posterior density $\pi^*$ is calculated as

$$\pi^*(m,\alpha,k|\vec{x}) = \frac{l(\vec{\mathbf{x}}|m,\alpha,k)\pi(m,\alpha,k)}{\iiint \; l(\vec{\mathbf{x}}|m,\alpha,k)\pi(m,\alpha,k)dm\,d\alpha\,dk} \qquad (2)$$

Where $\pi$ is the prior and $l(\vec{\mathbf{x}}|m,\alpha,k)$ is the likelihood of the observation vector $\vec{\mathbf{x}}$ given the parameters $m,\alpha,k$. The denominator makes the integral under the pdf equal one.

We used non-informative priors for the location and scale parameters, (i.e. the location parameter and the log-transformed scale parameter were uniform). A normal distribution with standard deviation 0.2 and expectation 0.0 was used as prior for the shape parameter $k$, inspired by Coles and Dixon (1999), Martins and Stedinger (2000), and Renard et al. (2013) .

The likelihood for the systematic data is (see Gaál et al., 2010; Stedinger and Cohn, 1986):

$$l_s = \prod_{i=1}^{n} f(x_i|m,\alpha,k) \qquad (3)$$

Where $f(x_i)$ is the probability density function for the GEV distribution with the parameter values $m,\alpha,k$ evaluated for the observation $x_i$. For historical- and paleofloods, it is assumed that all $g_j$ floods must exceed a threshold $x_{0,j}$ for the period $j$ where duration $h_j$ is known. The likelihood of $h_j$-$g_j$ number of floods not exceeding $x_{0,j}$ during the period $h_j$ is given as:

$$l_{b,j} = \left[F(x_{0,j}|m,\alpha,k)\right]^{h_j-g_j} \qquad (4)$$

Where $F$ is the GEV distribution given in Eq. (1).

We also need to include available knowledge on floods exceeding $x_{0,j}$. In the simplest case we know only that $g_j$ floods exceeded $x_{0,j}$, if so likelihood can be written as:

$$l_{a1,j} = [1 - F(x_0|m,\alpha,k)]^{g_j} \qquad (5)$$

Alternatively, we might know that the floods that exceeded $x_0$ took place within an interval defined by an upper $x_U$ and lower $x_L$ limit:

$$l_{a2,j} = \prod_{o=1}^{g_j}\left[F(x_{U,o}|m,\alpha,k) - F(x_{L,o}|m,\alpha,k)\right] \qquad (6)$$

And, in optimal scenario, we know the exact magnitude of all floods exceeding $x_{0,j}$:

$$l_{a3,j} = \prod_{o=1}^{g_j} f(y_o|m,\alpha,k) \qquad (7)$$

The total likelihood is given as a product of the three major likelihood terms:

$$l_i = l_s \prod_{j=1}^{J} l_{ai,j} l_{b,j} \qquad (8)$$

Where J is the number of sub-periods with specific perception thresholds.

The posterior distribution of the parameters was estimated using a MCMC-method implemented in the R-package nsRFA (Viglione, 2012). For estimating return levels, we used the posterior modal values of the parameters. It poses a





challenge to set the perception threshold $x_0$ and length of the historical floods $h$, i.e. for which period the listed floods represents
all floods above the threshold. A simple rule is to set the perception threshold to the lowest observed historical flood value in
the historical period. The length of the historical period was decided using the average spacing approach as recommended by
Engeland et al. (2018) and Prosdocimi (2018).
*3.4.2 Non-stationary flood frequency modelling*
We applied a simple approach to get an estimate of the non-stationary 200-year flood during the recent 1000 year using the
paleorecord. In a first step the parameters m', α' and k' in the GEV distribution were estimated using the systematic flood
observations. Then we can estimate flood quantiles as:

$$x(F|m',\alpha',k') = \begin{cases} m' + \frac{\alpha'}{k'}\left[1 - (1 - ln(F))^{k'}\right] & k' \neq 0 \\ m' - \alpha'[ln(-ln(F))] & k' = 0 \end{cases}$$
   (9)

Note that by replacing $F$ with $1-1/T$ in Eq. (9) we can calculate the flood quantiles for the return period T.
From the sediment core we can estimate a time series of the probability of exceedance $w_t$ of the threshold $u$, for each year $t$ if
we calculate the exceedance rates $w_t$ as the mean number of excesses in a sufficiently large moving window. If we assume that
the observed non-stationary exceedance rate influences both the location and scale parameters with a common factor $r_t$ , we
see from Eq. (9) that
$x(F = 1 - w_t|r_t m', r_t \alpha', k') = r_t x(F = 1 - w_t|m',\alpha',k') = v$    (10)
Since the threshold $v$, and the exceedance rate $w_t$ is known, the factor $r_t$ can be estimated as:
$r_t = v/x(F = 1 - w_t|m',\alpha',k')$    (11)
The T-years flood for the time t can then be estimated as:
$q_{Tt} = r_t x(F = 1 - 1/T|m',\alpha',k')$    (12)
**4 Results**
**4.1 Flood variability from the lake sediment cores**
The shortest core (FLS113) is 18 cm long, and represent the period AD 1953-2013. The longest core (FLP213) is 516 cm long
and represents the period approximately 0-10 300 years before present (present = 1950).
FLP213
The results from the XRF-scan (Ti$_{/total\ scatter}$, Ca$_{/total\ scatter}$ and K$_{/total\ scatter}$) and the greyscale-value (BW) from photo of the core
are shown as a function of depth in Fig. 9 together with a photo of FLP213. The core consists of a dark brown gyttja with
preserved macro fossils including leaf fragments. This gyttja, carrying a low minerogenic content, is referred to here as 'the
background signal' which is characterized by its dark color (BW<30), high LOI (30-40%), low DBD (<0.3 g/cm$^3$) and
magnetic susceptibility (MS) with values close to zero (<5 SI*10$^{-5}$). Moreover, it returns low K$_{/total\ scatter}$ (<0.03), Ti$_{/total\ scatter}$
(<0.03) and Ca$_{/total\ scatter}$ (<0.03). Interspersed in this 'organic slush' there are narrow (mm scale) light grey (BW 40-170)
minerogenic layers with LOI lower than 20%, relatively high density (DBD 0.5-1.0 g/cm$^3$),  higher than average MS with
peaks at 15-20 SI*10$^{-5}$ as well as peaks in K$_{/total\ scatter}$ (0.1-0.9), Ti$_{/total\ scatter}$ (0.1-0.4) and Ca$_{/total\ scatter}$ (0.1-0.7). At 33.5-18.0 cm
depth in core there is an anomalous thick minerogenic layer with LOI at <2%, DBD at1.6 (g/cm$^3$), MS at 98 SI*10$^{-5}$, and very
high K$_{/total\ scatter}$ (0.6), Ti$_{/total\ scatter}$ (0.4) and Ca$_{/total\ scatter}$ (0.7).

38        The correlation matrix (Table 3) shows strong (and significant) correlations between K$_{/total\ scatter}$, Ti$_{/total\ scatter}$, Ca$_{/total}$

$_{scatter}$, MS and BW. The weakest correlation is 0.74 between MS and BW which is still very high. LOI is, as expected, negatively



correlated with all the other measured variables. We suggest that the main process explaining the relationships between these
parameters is driven by the on-off signal related to transport of minerogenic material to Flyginnsjøen during bifurcation events.

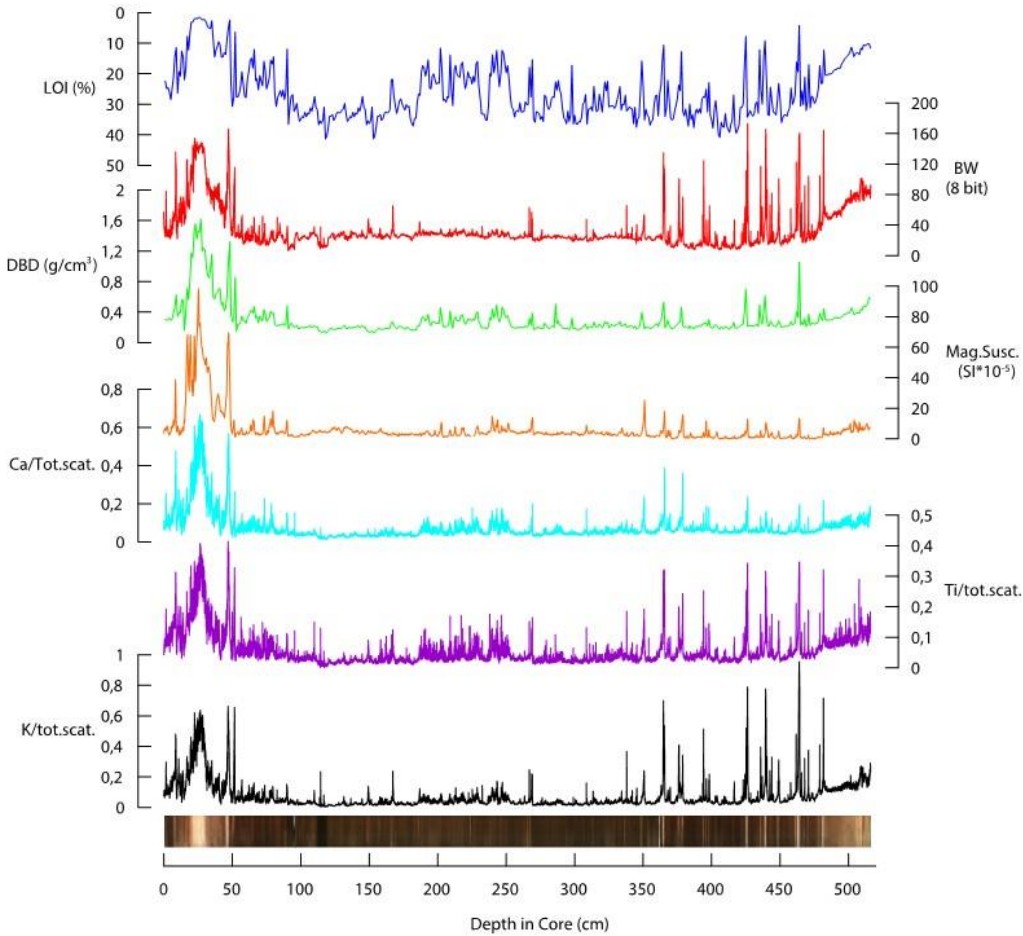

Figure 9: Results from measured parameters in FLP213. Loss-on-ignition (LOI %) indicated content of organic matter in the
core, and are plotted on an inverse scale (blue). BW (red) shows the 8-bit (0-255) black-white values extracted from a photo
of the core surface where 0 is black. Dry bulk density (DBD) is plotted in unit gram per cm$^3$ (green). Magnetic susceptibility
(orange) is plotted as SI*10$^{-5}$ as magnetic susceptibility is a dimensionless parameter. XRF-data (K, Ca and Ti) are normalized
against total scatter to reduce potential effect of water content.



Table 3 Correlation between measured parameters in FLP213/FLS113. LOI ( %) indicate content of organic matter in the core,
BW is the 8-bit (0-255) black-white values extracted from a photo of the core surface where 0 is black. CT grayscale is a 16-
bit number indicate relative densities of the core, DBD is given in unit gram per $cm^3$ (green). MS is measured as $SI*10^{-5}$ (it is
a dimensionless parameter). XRF-data (K, Ca and Ti) are normalized against total scatter to reduce effect of water content. All
correlations are significantly different from zero.

| | LoI | BW | CT greyscale | DBD | MS | K/total scatter | Ca/total scatter | Ti//total scatter |
|---|---|---|---|---|---|---|---|---|
| **LoI** | 1 | -0.67 / - | - / - | -0.82 / - | -0.61 / - | -0.61/- | -0.64 / - | -0.67 /- |
| **BW** | -0.67 / - | 1 | - / - | 0.82 / - | 0.74 / 0.73 | 0.89 / - | 0.81 / | 0.89 / - |
| **CT greyscale** | | - / - | 1 | - / - | - / 0.79 | - / 0.64 | - / 0.68 | - / 0.59 |
| **DBD** | -0.82/ - | 0.82 / - | - / - | 1 | 0.86 / - | 0.77 /- | 0.87 /- | 0.82 / - |
| **MS** | -0.61 / - | 0.74 / 0.73 | - / 0.79 | 0.86 / - | 1 | 0.76 / 0.66 | 0.86 / 0.73 | 0.76 / 0.63 |
| **K/total scatter** | -0.61 / - | 0.89 / - | - / 0.64 | 0.77 / - | 0.76 / 0.66 | 1 | 0.85/ 0.93 | 0.96 / 0.95 |
| **Ca/total scatter** | -0.64 / - | 0.81 / - | - / 0.68 | 0.87 / - | 0.86 / 0.73 | 0.85 / 0.93 | 1 | 0.91 / 0.88 |
| **Ti//total scatter** | -0.67 / - | 0.89 / - | - / 0.59 | 0.82 / - | 0.76 / 0.63 | 0.96 / 0.95 | 0.91 / 0.88 | 1 |

FLS113
This core shows dark organic gyttja with light grey minerogenic layers, similarly to FLS213. The minerogenic layers yield
high values of $K_{/total\ scatter}$ (0.2-0.8), $Ca_{/total\ scatter}$ (0.1-0.4) and $Ti_{//total\ scatter}$ (0.1-0.2) as well as slight increase in MS ($>6\ SI*10^{-5}$)
(Fig. 10). CT data shows that the light grey layers are of high density and reveals numerous thinner layers not visible on photo
or in the lower-resolution XRF and MS data. Slight offsets in the positioning of layers in the CT imagery and optical photo
occurs due to the fact that the layering is not entirely horizontal.

15       Correlation coefficients between CT greyscale values, MS, $K_{/total\ scatter}$, $Ca_{/total\ scatter}$ and $Ti_{/total\ scatter}$ in FLS 113 are all

over 0.59 and significantly larger than zero. The strongest correlation is seen between $K_{/total\ scatter}$, $Ca_{/total\ scatter}$ and $Ti_{/total\ scatter}$
(Table 3). The somewhat weaker correlation to MS and CT greyscale, and the fact that CT imagery show layering (e.g 11-12
cm depth in core) not picked up by the other data (Fig. 10), can partly be explained by slight offsets in the positioning of layers
between the different scans as well as differences in sampling resolution. The strong correlations and general picture of layered
intervals yielding high values, however, indicates that one dominating factor 'controls' the variability, providing further
support of the interpretation that transport of minerogenic material to Flyginnsjøen during bifurcation events is the main
process.

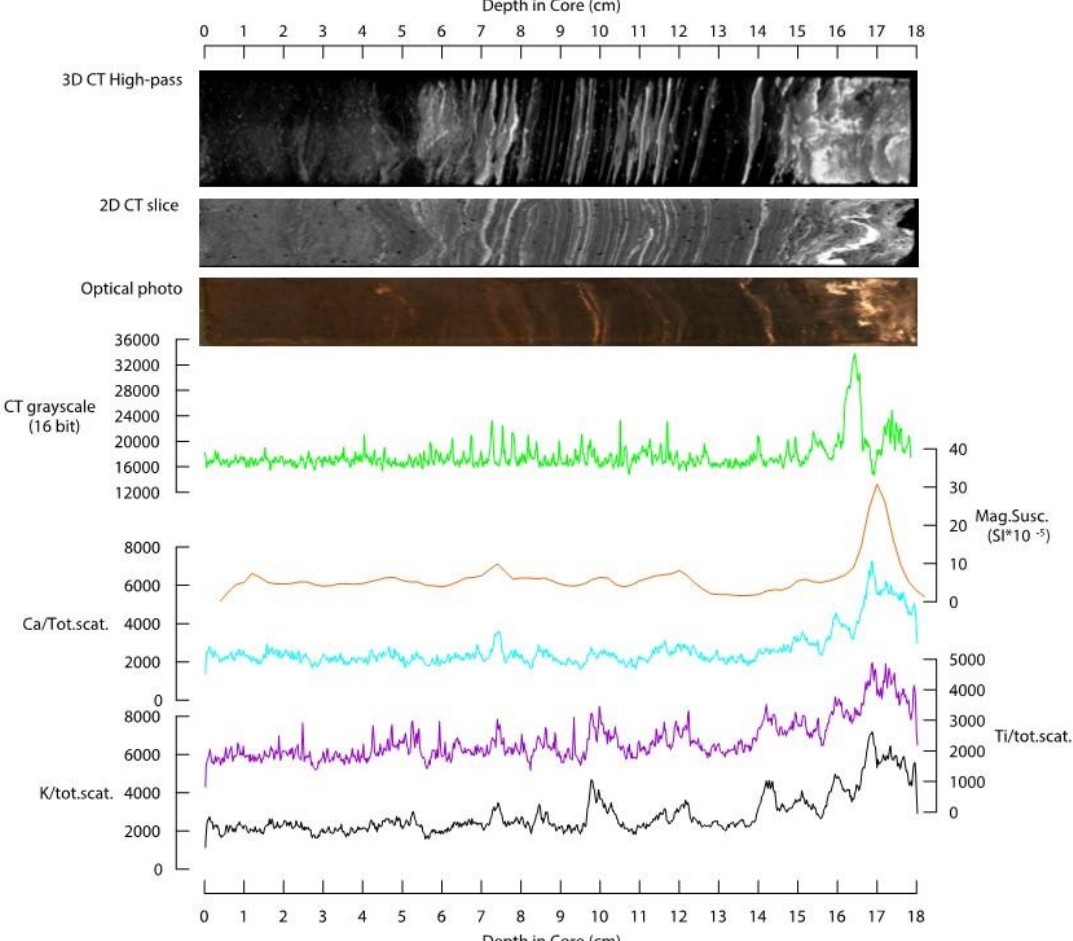

Figure 10: Results from high resolution analysis of core FLS113. The top panel shows a 3D CT-visualization of high-density
layers (white) in the core. The 2D slice is an 80μm thick slice from the middle of the sediment core. The Optical photo is an
RGB photo of the surface of the halved sediment core. CT grayscale plot (green) shows an 80μm grayscale variability along
a line through the middle of the sediment core. MS (orange) is plotted as $SI*10^{-5}$ as magnetic susceptibility is a dimensionless
parameter. XRF-data (K, Ca and Ti) are normalized against total scatter to reduce effect of water content.
**4.1.1 Age-depth models**
To establish an age-depth relationship for the cores, sediments were subjected to lead dating $^{210}$Pb (FLS113) and radiocarbon
dating ($^{14}$C) of FLP213. Measurements were performed by the Environmental Radioactive Research Center at the University
of Liverpool (Appleby and Piliposian, 2014) and Poland (Poznan Radiocarbon Laboratory) (Goslar, 2014). The $^{210}$Pb and 14C
dates used to establish the age-depth models presented in Fig. 11 are listed in Table 4 and 5. Estimation of age as a function
of depth for FLS113 was done using a quadratic term regression model of CRS model calculations of the $^{210}$Pb with the 1963
$^{137}$Cs peak at 16.25 cm depth in core (Table 4) as a reference point (Appleby, 2001). For FLP213, we used a Bacon age-depth
modelling approach (Blaauw and Christeny, 2011) available in the R-package Bacon. One $^{14}$C sample from 51 cm depth in
FLP213 was rejected, as this has a stratigraphically reversed age (see Table 5). The age is clearly too old, possibly related to



high content of saw dust bringing in relative old carbon core at depth in core. The saw dust may have originated from a saw
mill in the catchment at this time. The 15.5 cm thick anomalous layer at 18.0-33.5 cm depth in core was classified as "slump"
in the Bacon model, and thus interpreted as an instantaneous event deposit. This layer has a basal age estimate of 1776 CE
from the age-model, and is likely to be related to the historically documented 1789 CE Stor-Ofsen flood event (see Sect. 3.2).
Table 4: Fallout radionuclide concentrations and chronology for FLS113 from Flyginnsjøen.

| Depth (cm) | $^{210}Pb_{Total}$ (Bq kg$^{-1}$) | ± | $^{210}Pb_{Unsupp.}$ (Bq kg$^{-1}$) | ± | $^{210}Pb_{Supp.}$ (Bq kg$^{-1}$) | ± | $^{137}Cs$ (Bq kg$^{-1}$) | ± | Year | Uncertainty (years) |
|---|---|---|---|---|---|---|---|---|---|---|
| 0 | | | | | | | | | 2013 | 1 |
| 0.25 | 809.5 | 47.9 | 702.3 | 49.2 | 107.2 | 11.3 | 65.7 | 7.2 | 2013 | 1 |
| 1.25 | 686.2 | 33.4 | 585.9 | 34.0 | 100.3 | 6.6 | 63.3 | 5.3 | 2011 | 1 |
| 2.25 | 570.9 | 21.6 | 492.4 | 21.9 | 78.5 | 3.9 | 62.0 | 3.4 | 2009 | 1 |
| 3.25 | 598.8 | 22.6 | 524.3 | 23.0 | 74.5 | 4.2 | 72.7 | 3.7 | 2007 | 2 |
| 4.25 | 549.2 | 21.5 | 474.9 | 21.9 | 74.2 | 3.9 | 82.9 | 4.3 | 2004 | 2 |
| 5.25 | 455.9 | 17.5 | 386.0 | 17.8 | 69.8 | 3.1 | 77.6 | 3.4 | 2000 | 2 |
| 6.25 | 482.0 | 25.2 | 404.0 | 25.6 | 78.0 | 4.7 | 64.0 | 3.9 | 1998 | 2 |
| 8.25 | 515.6 | 20.4 | 442.3 | 20.7 | 73.2 | 3.7 | 58.9 | 3.3 | 1992 | 3 |
| 10.25 | 391.4 | 19.3 | 329.6 | 19.6 | 61.8 | 3.6 | 84.6 | 3.7 | 1986 | 3 |
| 12.25 | 331.6 | 15.3 | 266.2 | 15.6 | 65.4 | 3.0 | 78.1 | 3.1 | 1979 | 4 |
| 14.25 | 231.2 | 12.8 | 173.4 | 13.1 | 57.9 | 2.6 | 68.0 | 2.8 | 1970 | 5 |
| 16.25 | 226.4 | 13.8 | 152.8 | 14.1 | 73.7 | 3.1 | 138.8 | 4.1 | 1962 | 6 |
| 17.25 | 193.3 | 13.3 | 140.7 | 13.5 | 52.6 | 2.7 | 50.7 | 2.4 | 1957 | 6 |
| 19.25 | 112.9 | 7.3 | 68.8 | 7.4 | 44.1 | 1.6 | 9.2 | 1.2 | 1948 | 7 |

Table 5: $^{14}$C-dates for FLP213 from Flyginnsjøen. Radiocarbon ages are calibrated using the IntCal 13 calibration curve
(Reimer et al., 2013)

| Lab. Nr. | Depth in core (cm) | $^{14}$C age. yr BP | Cal. yr BP (most prob. 68.3% conf int.) |
|---|---|---|---|
| Poz-57974 | 51 | 870 ± 30 | 732 –796 (0.97) |
| Poz-59030 | 70 | 390± 30 | 453 – 503 (0.78) |
| Poz-57975 | 118 | 1565± 35 | 1455 – 1521 (0.73) |
| Poz-57976 | 206 | 2860± 40 | 2924 – 3037 (0.91) |
| Poz-57977 | 304 | 4125± 40 | 4571 – 4653 (0.49) |
| Poz-57978 | 370 | 5670± 40 | 6409 – 6487 (1.00) |
| Poz-59029 | 401 | 6535± 35 | 7424 – 7476 (1.00) |
| Poz-57979 | 462 | 8180± 50 | 9028 – 9140 (0.75) |
| Poz-57980 | 504 | 9190± 50 | 10259 – 10403 (1.00) |

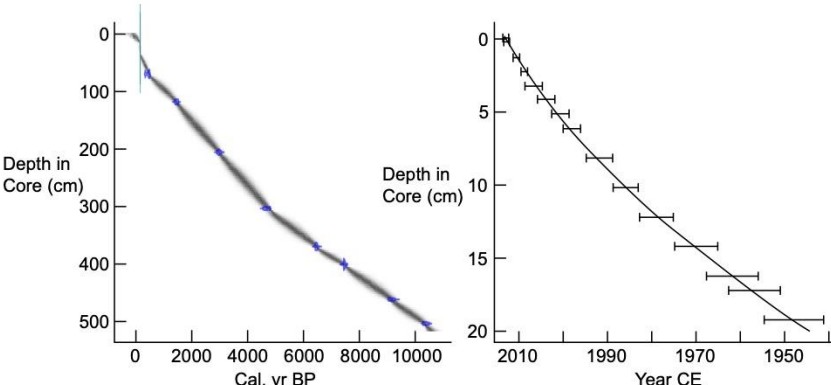

Figure 11 Age-depth model for FLP213 (right) and FLS113 (left). Note the step in the FLP213 age-depth model at 33.5 – 18.0
cm depth in core related to the *Stor-Ofsen* flood event in 1789 CE.



### 4.1.3 Identification of flood layers in FLS113.

We used the concentration of Ti$_{/total scatter}$ and K$_{/total scatter}$ from the XRF-scan of FLS113 to establish a link between dense, minerogenic sediment layers and the 22 bifurcation events between 1953 and 2013. Note that XRF data (K$_{/total scatter}$, Ca$_{/total scatter}$ and Ti$_{/total scatter}$) correlates strongly with the CT-scan (greyscale values), and MS for both FLS113 and FLP213 (Table 3), and this suggests the flood transported material originate from a one source and that this is constant over time. All detected layers are thus interpreted to be related to the same process bringing minerogenic material to Flyginnsjøen. The first step in our approach was to transform the depth of the XRF-scan to age using the depth-age model for FLS113. After having identified the flood layers, we used the algorithm described in Sect. 3.3.1 to identify local peaks in the measured parameter. We used a time window of 1 year, a value of 680 and 527 for Ti$_{/total scatter}$ and K$_{/total scatter}$ respectively for $h_1$ and $h_2=0.5* h_1$ which identified 23 local peaks for Ti$_{/total scatter}$ and K$_{/total scatter}$ over the same period that we observe 22 bifurcation events. A time series of the bifurcation volumes and the XRF-scan data can be viewed in Fig. 12. Taking into account the uncertainty in the dating (Fig. 11), we see that five of the bifurcation events do not correspond directly to a sediment layer. All the three largest flood events were, however, correctly identified, and considering the uncertainties in the age-depth model this supports our working hypothesis that sediment layers can be used to identify flood events caused by episodes of bifurcation at Kongsvinger.

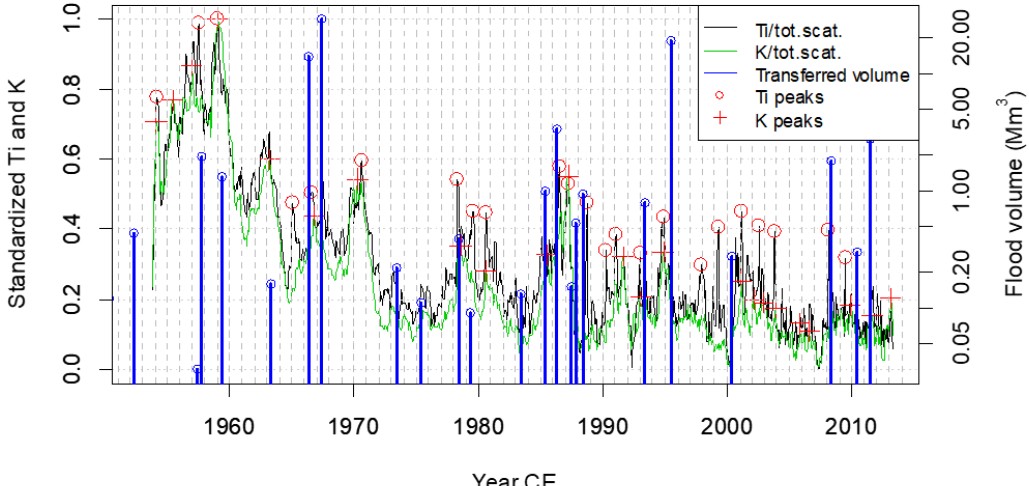

Figure 12. Transferred volume of the 23 bifurcation events in the period 1950-2013 CE (in blue) and the 24 identified flood-layers (red) identified using XRF scans of Ti$_{/total scatter}$ and K$_{/total scatter}$ for FLS113.

### 4.1.3 Frequency of flood events during the Holocene.

From FLS113 we have established a link between dense, minerogenic sediment layers and bifurcation events. We therefore assumed that the analyses of FLP213 can be used to produce a time series of flood events covering the last 10 300 years. Here we have used the local peak detection algorithm presented above to identify sediment layers with high concentration of K$_{/total scatter}$ and Ti$_{/total scatter}$. Since the uncertainty range in the age estimate is 30 to 50 years, we calculate the average rate of a given flood event within a moving Gaussian time windows of 50 years for both Ti$_{/total scatter}$ and K$_{/total scatter}$ (Fig. 13). The standard deviation of the estimated flood rate $\hat{\lambda}$ was calculated as $\hat{\lambda} \pm z\sqrt{\frac{\hat{\lambda}(1-\hat{\lambda})}{50}}$ and it was used to assess the 95% confidence intervals.





We see that the flood counts using $Ti_{total\ scatter}$, $K_{total\ scatter}$ and BW to a large degree overlap and follow the same Holocene
trends, as anticipated due to the high correlation coefficient between the two (see above).

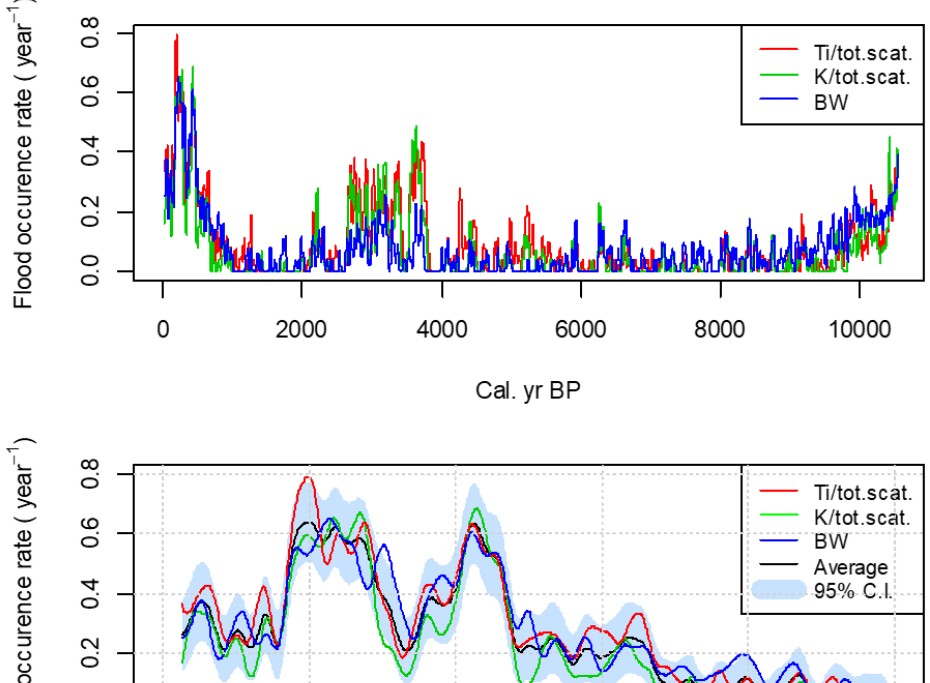

Figure 13: In the top panel, average flood rate per year calculated in a 50-years moving window during the Holocene. In the
lower panel, the recent 1000 years are shown only. The lower panel also include a 95% confidence interval for the average
flood rates. The flood rates were identified by detecting local peaks in $Ti_{total\ scatter}$, $K_{total\ scatter}$ and BW values.
**4.2 Stationarity of flood frequency in the paleo-flood data**
A key observation in the Holocene flood frequency reconstruction is the large non-stationarity played out across multiple time
scales.  We observe that there are two major flood rich periods during the Holocene (Fig. 13, upper panel). The first runs from
3800 to 2000 cal. yr BP when it ends abruptly. The second period extend from around 600 cal. yr BP up to present day. Looking
at flood frequency over the recent 1000 years (Fig. 13, lower panel) we observe significant internal variability within the flood
rich period. The period with the highest flood rates occurs in the 18[th] century, but also in the 15[th] century. The high flood
frequency in the 18[th] century is also recorded in the historical flood data (Fig. 6). The data from FLP213 informs us that the
flood event in 1789 is truly an anomaly, as is evident from the sheer amount of sediments deposited during this event (no other
flood comes close), and it also yield the highest measured values of e.g. density (DBD) as well as magnetic susceptibility (MS)
throughout the core (Fig. 9). It is therefore reasonable to assume that the 1789 CE flood was an extraordinary event making it
the largest during the entire time span of the record, i.e. 10 300 years.





**4.3 Flood quantile estimation by combining systematic-, historical- and paleo-flood data**
The flood quantiles combining the systematic, historical and paleodata have been analysed in different, but complementary
ways. The first step in this approach is to estimate the flood quantiles using only systematic data whereupon we included all
the historical data. The length of the historical data period was calculated based on Prosdocimi (2018) and Engeland et al.
(2018). The smallest historical flood of 2533 m$^3$/s was used as the threshold $x_0$. The average waiting time between the historical
flood is 22 years for the historical period that started in 1653 CE and ended in 1872 CE giving $h = 219$ years. The exact sizes
of the historical floods (Table 1) was assumed. In the third approach we used the paleo-record as a guidance to weigh the
historical information. Since paleorecord indicates that the historical floods in the 18$^{th}$ century occurred in a flood rich period,
we used only the historical flood events from the 19$^{th}$ century. Moreover, the historical flood from 1789 CE was included, and
it was suggested that this was the largest flood during the last 10 000 years for reasons explained above. The results are shown
in Fig. 14, and we see that the results are sensitive to the assumption of which period the 1789 CE flood represents.

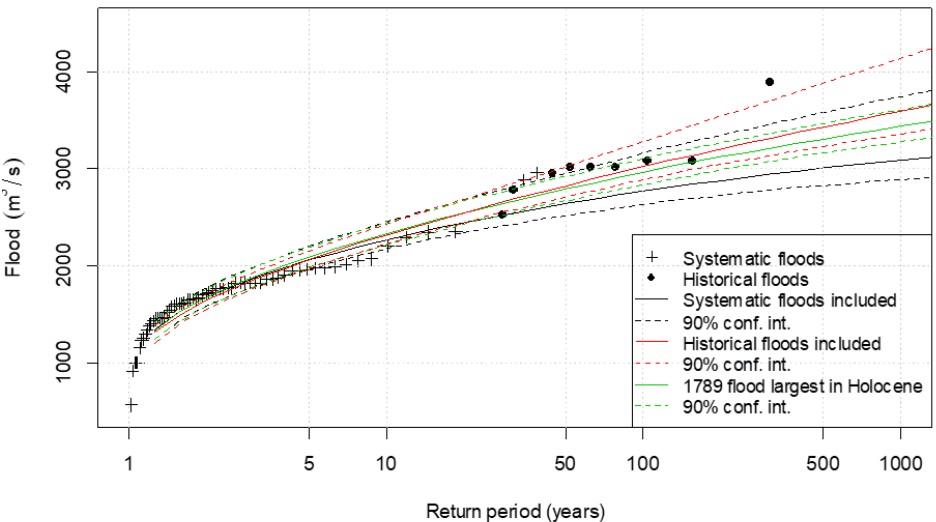

Figure 14: The sensitivity of flood frequency analysis to historical floods.
The next step was to include the paleo-flood information in the flood frequency analysis. We did this in two ways: (i) we
combined the systematic data and the paleodata and (ii) we combined systematic, historical and paleodata. For the paleodata
we used 1800 m$^3$/s as the threshold $x_0$ since it provided the same number of flood events (i.e. 19 events) from the paleo record
and the streamflow observations for the overlapping time period (1891-1950). In case (ii) above, we counted 91 flood events
representing a period of 330 years (1320-1650 CE), and for case (iii), we counted 179 events for a period of 540 years (1320-
1850 CE) The results are shown in Fig. 15. We see that the estimates are sensitive to historical information. The paleodata did
not impact the result to the same degree.





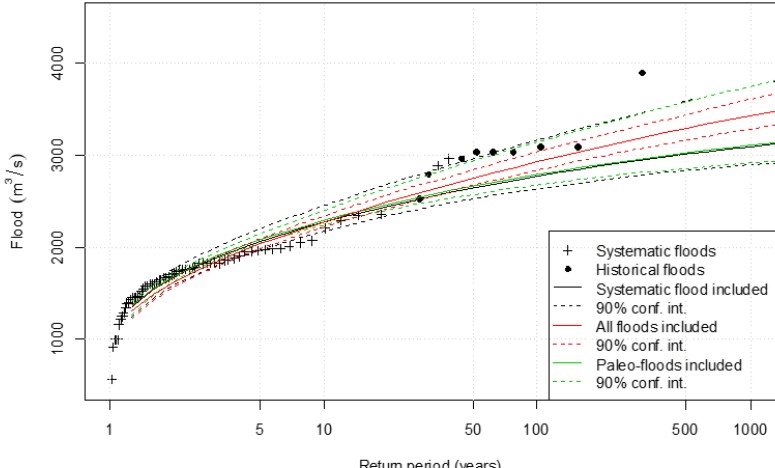

Figure 15. The sensitivity of flood frequency analysis using paleoflood data.
To achieve a nonstationary estimate of the design flood, we used the flood occurrence rate presented in Fig. 13 to estimate the
200-years flood in a moving time window as explained in Sect. 3.4.2. We used 1900 m$^3$/s as the threshold $v$ in Eq. (11) since
it provided a good agreement between the 200-years flood estimated from the systematic data and the non-stationary 200-years
flood for the overlapping period. The results are presented in Fig. 16. We now see that the size of the 200-years flood is non-
stationary. During the 'Little Ice Age' (LIA) it was up to 23% higher than in present climate, whereas during the period 4000-
6000 B.P it was around 30% lower than today.

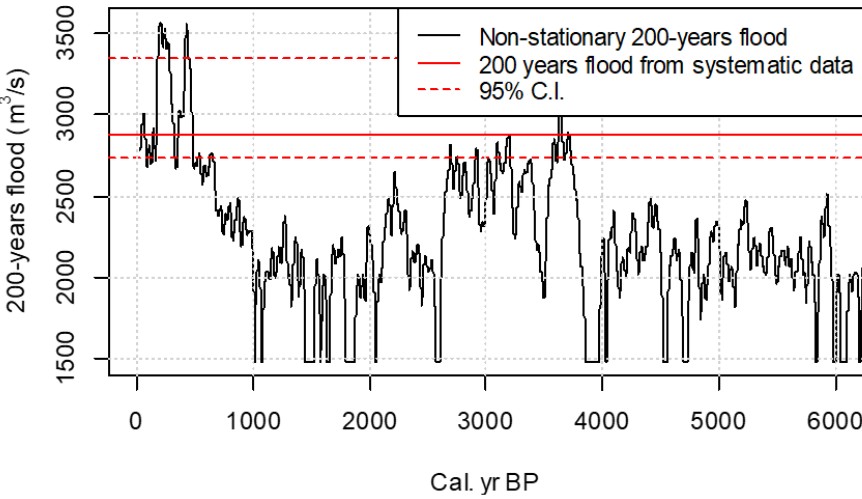

Figure 16: Non-stationary estimate of the 200-years flood for the resent 6000 years. The red lines indicate the estimated 200-
years flood and the 95% confidence intervals estimated using systematic streamflow observations.



**5.0 Discussion**
**5.1 The reliability of the paleoflood records**
During the last decade or so lakes across Europe have been studied in detail and high-resolution paleoflood records have been
produced from both the low-lands and the high-lands (Wilhelm et al., 2018). Unlike many of these studies, we have worked
with lakes that *only* receive flood-delivered sediments whenever the local river (Glomma) exceeds a certain well-known
threshold (1500 m$^3$/s). This setting tends to suggest that not only are we working with a sedimentary archive that filters out
noise, but also one that provides a minimum estimate of the discharge associated with the floods recorded. The flood
information extracted from the lake sediment cores, nevertheless, relies on a set of assumptions that is discussed in the
following.
The first assumption is that all flood events recorded in lake Flyginnsjøen are directly related to Glomma. We cannot
completely rule out the possibility that minor floods in the local catchment of Flyginnsjøen occurred simultaneously with
floods originating from Glomma or even just within the very small catchment surrounding the lake due to local rainstorm
events. Given the heavy vegetation cover in the catchment of Flyginnsjøen, its small size and the low-angles of the slopes
leading into the lake, we deem the possibility of a local sedimentary imprint as very low. This is supported by both XRF and
MS data. The consistency in bifurcation events causing peaks in concentration in both Ti$_{/total\ scatter}$ and K$_{/total\ scatter}$, as well as
MS, suggests that the source region for this signal remains the same throughout the record. The most likely source is thus the
abundant glaciofluvial material available in the area between Tarven and Flyginnsjøen (see Fig. 4).
A second assumption is that is that the river channel and landscape geometry controlling the bifurcation events has
not changed over the approximately recent 10 000 years to the extent that it alters this interplay between a flooding Glomma
and the investigated lake. The current river geometry was shaped by a glacial lake outburst flood (GLOF) some 10 500 years
ago with a peak discharge of more than 10$^6$ m$^3$/s (Høgaas and Longva, 2016). This GLOF flushed the valley where Glomma
runs and also established the current river channel at Kongsvinger (Pettersson, 2000). Based on (Klæboe, 1946) and (Hegge,
1968)the threshold between Vingersjøen and Flyginnsjøen (Fig. 4) is a resilient and stable topographic feature. The intermittent
drainage patterns that route water from Vingersjøen to Flyginnsjøen during the bifurcation events may have undergone some
changes during the course of time, but it's hard to see how this would directly influence the deposition of flood-delivered
sediments to Flyginnsjøen. According to (Hegge, 1968), the flood events that occurred in 1967CE and 1968 CE caused some
erosion at the very highest elevation of this intermittent water course. Having said that, these flood events did not cause any
major damages to this area (Klæboe, 1946). In recent years, denser vegetation and also the construction of a road bridge has
potentially lessened the transfer capacity between the lakes although we have little or no evidence for this based on what we
observe in the lake core.
The resolution of the XRF signal is on average sub-annually, but because the uncertainty in the age-depth we
calculated flood rates i.e. average number of flood events for a moving 50 years window. Although the floods are of varying
magnitude, there appears to be no systematic relationship between, for instance, sediment thickness and flood sizes with the
exception of *Stor-Ofsen*. This is probably explained by the fact that the sediment transport for individual floods will in part be
deposited in the two preceding lakes (Vingersjøen and Tarven) buffering Flyginnsjøen (Fig. 4), but may also indicate that
event-specific features such as ground frost or snow cover may regulate sediment availability.

**5.2 Non-stationarity in flood records and reginal climate co-variability**
The paleoflood data presented here document that the flood frequency is non-stationary during the last 10 300 years being
manifested on multiple time scales (Fig. 13). Non-stationarity is typically identified as quasi cyclic flood-rich and flood-poor
periods (for European studies, see e.g. Brázdil et al., 2005; Glaser et al., 2010; Hall et al., 2014; Jacobeit et al., 2003;





Kundzewicz and International Association of Hydrological Sciences., 2012; Mudelsee et al., 2004; Swierczynski et al., 2013)
where the flood rich period may last for 50-60 years (e.g. Glaser et al., 2010).

3      Comparing centennial to decadal scale variability in the flood frequency reconstruction from Flyginnsjøen with

regional summer temperature reconstructions (Fig. 17) and local records of glacier variability – which in Scandinavia is
primarily driven by summer temperatures and winter precipitation – we observe co-variability which may indicate that the
non-stationarity of flood frequency is related to non-stationarities in climate. The data from Flyginnsjøen shows, for instance,
two distinct intervals with high flood frequency during the 'Little Ice Age' (LIA), both played out on centennial time scales.
Since 1850 there's been a steady increase in summer temperature followed by a reduction in flood frequency. Enhanced
flooding during the LIA is observed in other lake studies from eastern Norway as well, including Atnasjø (Nesje et al., 2001),
Butjønna (Bøe et al., 2006), Meringdalsvannet (Støren et al., 2010) and also the river Grimsa in the headwater of Glomma
(Killingland, 2009).

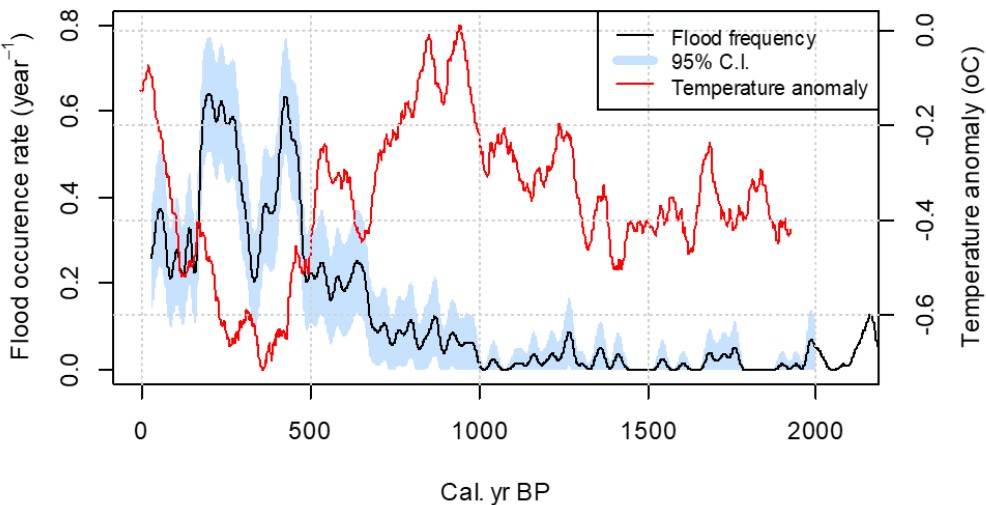

Figure 17: Flood frequency in Glomma (blue bars) and 30 years moving average Northern Hemisphere summer temperature
anomaly from Moberg et al (2015).
Another period with heightened flood activity occurs roughly between 4000 to 2000 years ago. The increase in flood frequency
in Glomma during this period, and also during the LIA interval, coincides with a recorded decrease in summer temperature at
Bruskardstjørni in eastern Jotunheimen (Velle et al., 2010) and increasing glacier growth in Rondane (Kvisvik et al., 2015),
the mountainous source area of Glomma (Fig. 18). Multi-decadal periods are typical superimposed on centennial trends, as is
the case for both these two flood rich intervals. The near absence of floods prior to 4000 years ago is another recurring feature
in all flood records from Eastern Norway (e.g., Støren et al., 2016). Locally, it seems plausible that the effect of raising the 0-
isotherm with 100-300 m altitude, the effect of a warmer summer, will significantly change the potential storage of snow
(Støren & Paasche, 2014).

25     Over the instrumental, and historic period floods in the Glomma catchment have occurred late in spring (late May,

early June) due to the presence of large snow reservoirs that suddenly starts to melt due to a rise in temperatures often combined
with persistent rain (Roald, 2013). The size of the spring flood depends on the total snow accumulation during winter, that is
controlled by both temperature and winter precipitation. Importantly, for these spring-snowmelt triggered floods, the soils are
either frozen and/or already saturated with moisture channeling most waters to or shallow sub-surface flow or overland flows





resulting in a fast response to meltwater and rain. The observed changes in flood frequency occurring both during the LIA and
in the first half of what sometimes is called the Neoglacial era (4000-2000 years ago) can thus, at least partially, be explained
by the combined effect of these flood generating processes (cf. Vormoor et al., 2016). The near-absence of floods prior to the
onset of the Neoglacial, when summer temperatures were ca 1°C higher than today (Velle et al., 2010), may be a valuable
albeit imperfect analogue for the coming century. During this period the 200-years flood is around 30% lower than today (Fig.

6    16).

7           In Lawrence (2020) a future climate in eastern Norway is suggest that smaller flood sizes might be expected in large

catchments where snow melt is the primary flood generating process. For small catchment, in western Norway, where rain-
generated floods already dominate, floods are expected to increase. Cooler temperatures, especially in summer and spring are
likely to delay the melting of the snow-cover which enhances the probability for a sudden warming because it occurs later in
the season.

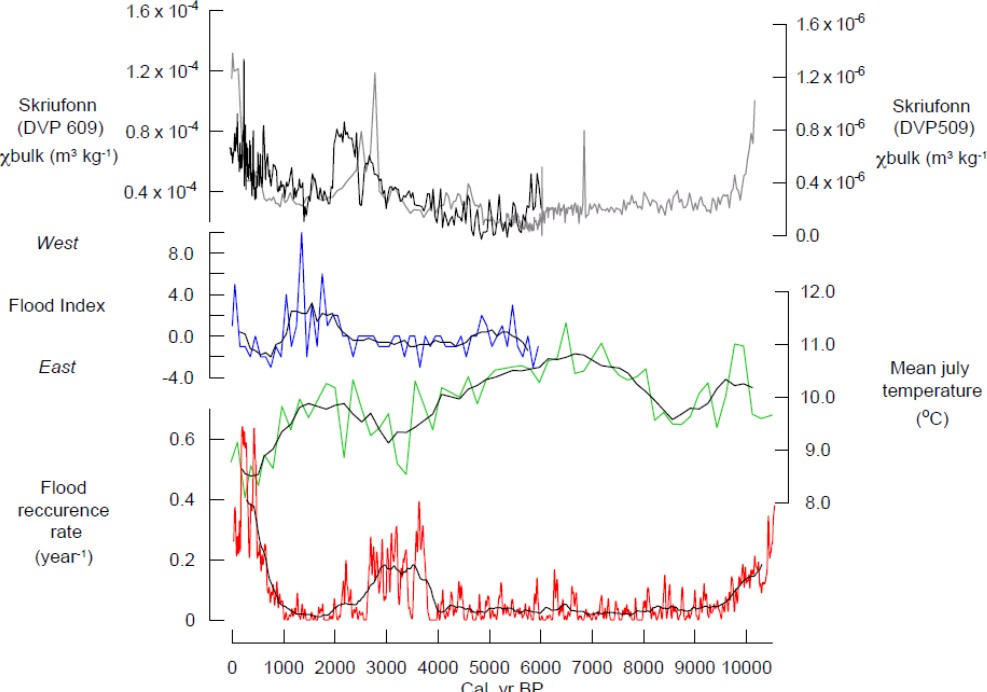

Figure 18: Flood frequency in Glomma (red) with 500yr running average, reconstructed summer temperature from
Brurskardtjørni, southern Norway (Velle et al 2010 (green) with 5-point running average,  Flood index (blue) with 5-point
running average (Støren et al., 2012) showing relative distribution of flood recurrence rate over southern Norway. Glacier
activity at Skriufonn, Rondane southern Norway (Kvisvik et al., 2015) (Black and purple)

18           The increase in flood frequency commencing at c. 4000 yr BP is a reoccurring feature not only in Europe but also in

parts of the USA (Paasche and Støren (2014). This hints at a large-scale change in the climate system at the time, with
implications for both atmospheric circulation patterns and temperature trends. This major climate shift recorded in Europe is
noteworthy because the flood seasonality is different across such a large area for many reasons, including the varying altitudinal
differences. In high-lying areas in Austria (north of the Alps, Swierczynski et al., 2013), and in the and the central alps
(Switzerland and northern Italy, Wirth et al., 2013) floods start to increase, as in eastern Norway, rapidly just after 4000 years
ago and remain on average high until 2000 years ago. Studying the relative distribution of floods in Norway, Støren et al.





(2012) suggest that the long-term trends in the floods is dependent on changes in the distribution of winter precipitation related
to semi-permanent shifts in atmospheric circulation patterns, and that an anomalous strong meridional component in the
atmospheric circulation pattern are linked to floods in eastern Norway. Over the time period between the two flood rich periods
in Glomma (c. 2000-1000 yr BP), Støren et al (2012) recorded a westward shift in the flood frequency likely caused by reduced
precipitation in the eastern areas.
There are also potential catchment feedback mechanisms, not necessarily related to climate, that can both dampen
and boost the flood patterns. Deforestation is, for instance, an additional explanation for the increase in flood frequency after
AD 1600. The mining industry that started in Norway in the 16[th] century required a large amount of timber which resulted in
widespread deforestation also in Glomma's catchment. Some of these mechanisms could potential help explain why *Stor-*
*Ofsen* in 1789 is the largest local flood on record. As mentioned above, the flood deposited the thickest sediment layer in the
entire record from Flyginnsjøen. The anomalous sediment thickness is also recorded in lake sediment archives for other places
in eastern Norway (see Bøe et al., 2006). Another amplifying process that can make floods become larger, and also
remobilizing lager amount of sediments, as was the case for *Stor-Ofsen*, was the large number of upstream landslides that took
place at the time (Roald, 2013). In fact, the summer of 1789 was named 'skriusommaren' (the landslide summer) in historical
material (Roald, 2013). We note that some of these historical slides might have occurred shortly after the flood as well.
**5.3 Flood quantile estimation by combining systematic-, historical- and paleoflood data**
The non-stationarity in flood frequency is a major challenge when estimating flood quantiles used for land planning and design
of infrastructure given that one needs to predict how the flood frequency will evolve over the life time of the construction, e.g.
for bridges it is 100 years (Koh et al., 2014). Milly et al. (2008) argued that 'stationarity is dead' and that it is necessary to
account for non-stationarity in order to avoid under-estimation of risks based on design floods.
Conversely, Serinaldi and Kilsby (2015) posited that 'stationarity is undead' because a stationary model is robust and
can be a useful reference/benchmark. Accounting for uncertainty in a stationary model can be as important as including non-
stationarity within a risk assessment framework. A non-stationary model introduces more parameters and thereby, in most
cases, increases the estimation uncertainty. An additional challenge when applying a non-stationary model for design flood
estimation is to project the flood frequency into the future.
The paleoflood data presented here suggests that the flood frequency is non-stationary and that there is indeed flood
rich and flood poor periods (Fig. 13). Since design flood estimates are used for assessing average risk over the lifetime of a
construction, it is desired that design flood estimates are stable over time and not sensitive to quasi cyclic variations in flood
sizes on annual to decadal time scales. It is, however, important to account for trends or shifts in flood frequency. Macdonald
et al. (2014) show that on centennial time scales, the effect of cyclic variations in short systematic records can effectively be
removed by a temporal extension of flood time series using historical information. Data from Flyginnsjøen and historical data
reveals that a quasi-stationary period can be identified at centennial time scales, but not on a sub-millennium time scale where
major shifts in flood frequency are identified (Fig. 13).
In this study, we firstly used the stationarity assumption and evaluated several possible ways to combine the three
data sources within a stationary framework. The results in Fig. 14 and 15 show that the design flood estimates are sensitive to
how we combine the systematic, historical and paleo flood data. We used 65 years of systematic data covering the period AD
1872 – 1936 for which we assume that the effect of river regulation is negligible. Adding the historical data from the flood
stone covering the period from AD 1675 to 1871, substantially increased the estimates of the flood quantiles and slightly
reduced the estimation uncertainty (Fig. 14).
The paleoflood timeseries provided here suggests that the flood frequency during the historical period is non-
stationary where the 18[th] century was an extremely flood rich period (Fig. 13), and that the AD 1789 flood was an exceptional





flood during the 10 300 years covered by the sediment core. Based on this paleo-information, we used historical data from the
19[th] century, and added the AD 1789 flood by assuming it was the largest flood over a period of 10 300 years. This slightly
reduced the flood quantile estimates as compared to using all historical information and substantially reduced the estimation
uncertainty (Fig. 14). These results shows that for the site at Elverum, we should be careful when including historical flood
information from the flood rich period in the 18[th] century.

6       As a next step, we added the paleo-flood data for the recent 600 years. This resulted in negligible differences in flood

quantile and uncertainty estimates (Fig. 15) indicating that the information content in the paleodata alone can be small. A
possible explanation is the combination of the relatively low threshold (according to Fig. 15 it is around a 5-years flood), and
that we only had information about the number of flood events. Both Macdonald et al. (2014)  and Engeland et al.  (2018)
show that the information content is low when the threshold for historical floods are too low.
In a final step we used the flood rate from the sediment core as a key to explore non-stationarity of the design flood
estimates, exemplified by the 200-years flood (Fig. 16). We could see important variation during the recent 6000 years. The
200 years flood was estimated to be around 23 % higher during the flood rich periods in the 18[th] century and 20% lower during
the warmest period. The high values for the 200-years flood during the 18[th] century is confirmed by the historical data. This
variation in design floods is, interestingly within the range seen in recent studies on climate change impacts on floods in
Norway (Lawrence, 2020). For a future climate that is expected to be warmer, the design flood might be expected to decrease.
Furthermore, this shows that the most interesting information we could get from the sediment core was the non-stationarity in
floods.

## 6.0 Conclusions

In this study we have (i) compiled historical flood data from existing literature, (ii) presented an analysis of sediment core
extracted from the lake Flyginnsjøen in Norway including results of XRF- and CT-scans plus MS measurements and used
these data to estimate flood frequency over a period of 10 300 years, and (iii) combined flood data from systematic streamflow
measurements, historical sources and lacustrine sediment cores for estimating design floods and assessing non-stationarities
in flood frequency at Elverum in the Glomma catchment located in eastern Norway. Our results show that

- Based on detailed analysis of lake sediments that trap sediments whenever the river Glomma exceeds a local threshold, we could estimate flood frequency in a moving window of 50 years throughout the last 10 300 years.

- The paleodata shows that the flood frequency is non-stationary across time scales. Flood rich periods has been identified, and these periods corresponds well to similar data in eastern Norway and also in the Alps such as the increase around 4000 years ago. The flood frequency can show significant non-stationarities within a flood rich period. The most recent period with a high flood frequency was the 18[th] century, and the 1789 flood (*Stor-Ofsen*) is probably the largest flood during the entire Holocene.

- The estimation of flood quantiles benefits from the use of historical and paleo data. The paleodata were in particular useful for evaluating the historical data. We identified that the 1789 flood was the largest one for the recent 10 300 years and that the 18[th] century was a flood rich period as compared to the 20[th] and 19[th] centuries. Using the frequency of floods obtained from the paleo-flood record resulted in minor changes in design flood estimates.

- We could use the paleodata to explore non-stationarity in design flood estimates. During the coldest period in the 18[th] century, the design flood was up to 23 % higher than today, and down to 30% lower in a warmer climate c. 4000-6000 years ago.





This study has demonstrated the usefulness of paleo-flood data and we suggest that paleodata has a high potential for detecting
links between climate dynamics and flood frequency. The data presented in this study could be used alone, or in combination
with paleo-flood data from other locations in Norway and Europe, to analyze the links between changes in climate and its
variability and flood frequency.
**Data availability**
Systematic flood data are available from the national hydrological database at the Norwegian Water Resources and Energy
Directorate. The data from the scanning of the sediment cores are available upon request to the authors.
**Author contribution**
The study was designed and planned by AA, KE, IS and ES. IS and ES carried out the lake coring and the field work. AA, IS.
ØP, and ES all contributed the scanning and analysis if the sediment cores. AA, KE and ES contributed to systematization of
historical and systematic flood data and the flood frequency analysis. AA prepared Figure 1 and 3 ES and ØP prepared Figure
4. ES prepared Figure 5 , 9, 10, 11 and 18. KE prepared Figure 2, 6, 7, 8, 12, 13, 14, 15, 16, and 17. KE prepared the manuscript
with contribution from all co-authors.
**Competing interests**
The authors declare that they have no conflict of interest.
**Acknowledgement**
We would like to extend our thanks to Svein Olaf Dahl, Nils Roar Sæltun and Chong-Yu Xu who co-supervised the master
projects of IS (Dept. of Geogarphy, UiB) and AA (Dept. of Geoscience, UiO) that create the basis for this study, and Karoline
Follestand and Martin Tvedt who assisted in coring lake Flyginnsjøen. This study became part of the Hordaflom project which
is funded by RFF-Vest, Hordaland County. ØP acknowledges the project ACER (812957), funded by Research Council of
Norway. All core samples, apart from the dating, were measured at the Earth Surface Sediment Laboratory (EARTHLAB)
(226171) at Department of Earth Science, University of Bergen.

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
