# Peer review of "New flood frequency estimates for the largest river in Norway based on the combination of short and long time series"

_Hydrology and Earth System Sciences, 2020_

## Referee Comment (RC1) · Anonymous Referee #1 · 20 Jul 2020

**New flood frequency estimates for the largest river in Norway based on the combination of short and long time series**

Kolbjørn Engeland, Anna Aano, Ida Steffensen, Eivind Støren, Øyvind Paasche

HESS

16/07/2020

This paper presents a very interesting case study for flood frequency analysis in the Glomma river in Norway, based on the combination of systematic, historical and paleoflood data. It benefits from a specific configuration of the Glomma river during large floods, when the flow level is exceeding a threshold. Flood deposits can be found within a small lake located 7 km from the main river.

It provides a valuable analysis of the variability of flood regime over the last 10 000 years. A flood rich period during the Little Ice Age does correspond to lower than average summer temperatures. As the future climate will be warmer, the design flood might decrease.

I have some recommendations that may improve the quality of the paper, which is already good.

**Length of the paleoflood record**

It is not clear how the authors state the value of 10 300 years from the sediment core. Is it the first dating of a sediment? If yes, according to the dating uncertainty (see a discussion in Dezileau et al., 2014, Geomorphology, 10.1016/j.geomorph.2014.03.017), it may be better to round it to 10 000 years. Another option is to refer to the huge glacial lake outburst flood (reported in section 5.1). It may be the beginning of the paleoflood record. Hogaas and Longva (2016) postulated a date of 10 000 – 10 400 years. So again, a rounded value of 10 000 years may be preferable.

**Section 1**

Page 2, line 3: the main reason for the increasing of flood damages is the increase of economic values within the flood plain. Impact of climate change is of concern, but do not forget that flood risk has two components: flood hazard and flood exposure.

**Section 2**

Figure 2 left: put one curve with dash lines (easier to follow)

Page 5, line 5: is the beginning of the Holocene period in Norway exactly 11 700 years ? Give a reference.

**Section 3**

Page 3, lines 8-11: it is not clear within section 3.1 which systematic period is considered. It is written 1871-1937, but in fact, floods larger than x0 = 2533 m3/s have been used (table 1: years 1966, 1967, 1995). If the flood regime was affected by river regulation after 1937, we expect to introduce a correction on the three largest floods after 1937. If not, you can consider the whole period 1871-2019. Please provide the p-value of a statistical test (e.g. Mann Kendall test) with the two periods: 1871-1937 and 1938-2018.

Page 10, lines 10-21: Stability of the geometry of the Glomma river is discussed in section 5.1. May be here in section 3.2.1 or in section 5.1, you could add information on the gaugings. Is the rating curve stable or do we need to use a set of different rating curves according to the gaugings?

**Section 4.3**

It is not easy to follow the different options of computation for flood frequency analysis:

- Systematic period: is it 1871-1937 (section 3.1) or 1872-1936 (section 5.3)?
- Historical period: 1653-1872. It may end in 1870 or 1871
- Paleoflood period: 1320-1850. Explain the starting date 1320. If paleoflood and historical floods are in agreement during the overlapping period, we expect to have 1320-1652

Plotting positions for figures 14 and 15 are not explained. According to Hirsch and Stedinger (1987), you should consider ALL the floods larger than the threshold x0 over the historical + systematic period, and then the systematic floods lower than x0. We expect to find a set of 14 peak flows (11 from historical period + 3 from the systematic period), and the remaining annual maximum values lower than x0 for the systematic period.

Plotting position of the 1789 flood should be different, according to option 1 or 2 (largest flood since 1653 or since 10 000 years).

On figure 5, it is not easy to understand which curves relate to cases (i), (ii) and (iii) vs systematic floods included, all floods included, paleoflood included.

**Section 5.2**

On figure 17, it is very interesting to see the flood rich period during the LIA with low temperature (and the contrary before 500 cal. yr BP). The authors could add a comment on the fact that we have two peaks during LIA but the temperature did not significantly change.

**Section 5.3**

Page 26, line 6: it remains unclear how the 600 year period is chosen. Additionally, according to section 4.3, it a rather a 530 year period (1320-1850) or a 332 year period (1320-1652)

**Small typos**

Page 2, line 18: they require that buildings

Page 2, line 22: up to 1000-year floods… (see Lovdata, 2010, and TEK17, 2018 for regulations

Page 3, lines 24-25: (Hanssen-Bauer et al., 2017

Page 5, line 16: the flow direction reverses

Page 5, line 24: in Fig. 5

Page 6, line 6: (after Hegge, 1968)

Page 10, line 4: reference of Ostmoe (1985) is missing in section 7

Figure 8: Legend "Max discharge"

Page 11, line 2: (Renberg and Hansson, 2008). Samples

Page 11, line 39 add "if k=0 (Gumbel distribution)" on the second part of equation (1)

Page 22, line 18: A second assumption is that the river

Page 22, line 20: Hogaas and Langva (2016) give an age of 10 – 10.4 cal ka BP in conclusion

Page 22, line 22: Based on Klaeboe (1946) and Hegge (1968) the threshold

Page 22, line 26: According to Hegge (1968), the… in 1967 CE

Page 22, line 38: and regional

Page 23, line 15: reference of Moberg et al. (2015) is missing in section 7

Page 24, line 14: Velle et al., 2010

Page 24, line 22: and in the central alps

Page 25, line 4: Storen et al. (2012)

Page 26, line 10: the threshold for historical floods is too low

Page 27, line 25: 7. References

---

## Referee Comment (RC2) · Daniel Schillereff (Referee) · 20 Jul 2020

**July 2020**

**Review of "New flood frequency estimates for the largest river in Norway based on the combination of short and long time series" by Engeland *et al*.**

**Manuscript ID: hess-2020-269**

**Review conducted by:**
**Dr Daniel Schillereff**
**Department of Geography, King's College London**
**daniel.schillereff@kcl.ac.uk**

This manuscript reports a Holocene palaeoflood reconstruction that amalgamates sedimentary, historical and instrumental data to refine flood frequency estimates for the Glomma catchment. I was really pleased to read and review this submission because of its interest and importance. Palaeoflood research has made big strides in recent years and embedding these data into design flood estimates is the next big task – and this is precisely what the authors have achieved in this paper. The manuscript is well written, clear and most of the analysis is convincing. The amalgamation and systematic analysis of independent (at least in terms of recording method) hydrological datasets is particularly effective and commendable. There are a handful of moderately substantive areas of clarification that I suggest are necessary prior to publication. I think these will result in a clearer narrative for the broad journal audience and present a more convincing analysis. A few minor comments are also raised.

**Comment #1: Structure and content of the introduction**

Given the broad audience of HESS, and the likelihood that some (many?) readers may be more accustomed to studying recent datasets, I suggest the authors incorporate more detail on the palaeodata in the introduction. I specifically suggest the authors elaborate on the ways in which palaeo data can help address the "two reasons" outlined on Page 2, Lines 23-26. Whilst many readers will have a sound knowledge of return periods and design flood estimates, applying palaeo data to these processes – and the value of doing so - may be quite new.

Similarly, I think it would be useful for the authors to say more about the timescales involved: how far back in time can palaeohydrological data be obtained from lake sediments (page 2, lines 42-43) and historical records (page 3, lines 5-11)? This would give readers a platform of knowledge that will be helpful when they engage with the return period calculations later in the paper. Likewise, I think the objectives (Page 3, lines 39-43 and on to Page 4) could be a bit more specific and pay particular attention to the timescale of your analysis

I also query whether Page 3, lines 5-11 could be slimmed down. To keep the focus on your work, perhaps mention briefly that there are a range of historical archives but place particular emphasis on epigraphic sources – especially flood stones - as that is the sort of data used later in this study

**Comment #2: Human modifications to the landscape and flood stationarity**

I am pleased the authors mention the possibility that changes in land-use could play a role and, similarly, I was really pleased that the authors explicitly assess whether the fluvial system behaviour would have allowed bifurcation events throughout the Holocene (Page 22, Lines 18-30). I think there

is scope to go further in ruling out the possibility that changes in flood frequency are driven by rather than amplified by human activity.

Page 3, line 24: it is important to state here that non-stationarity needn't only be a response to climatic variability. Human alteration of the land surface can also create a non-stationary fluvial system.

Page 5, Lines 2-5 talk about 'noteworthy land-use changes during the last 400 years, and specifically the removal of woodland cover". This is returned to briefly on Page 25, Lines 6-15 but I think this needs a more critical and in-depth evaluation. The authors acknowledge, for example, the notable rise in flood frequency rises around 500 yr BP, which happens to coincide with the assertion on Page 25 Line 8 "The mining industry that started in Norway in the 16th century required a large amount of timber which resulted in widespread deforestation also in Glomma's catchment". How confident are the authors that widespread deforestation amplified the climate driver but was not a driver of sedimentological change in its own right?

Similarly, I think the interpretation of external drivers, especially through the 2500-4000 yr BP flood-rich period, would be strengthened considerably if human interference could be ruled out. In Britain, for example, a number of fluvial and palaeolimnological studies show widespread mobilisation of sediments at that time resulting from settlement expansion. I have no idea about the mid- to late-Holocene history of human occupation in southern Norway but presenting or referring to data ruling this out possibility would really strengthen the case.

**Comment #3: Evaluating the geochemical flood proxy**

Overall, I find the proxy reconstruction to be convincing. I do wonder whether it could be strengthened by providing some information on the geochemical composition of the glacio-fluvial material between the two lakes. The authors state the Glomma catchment is the largest in Norway. To what extent will sediments being deposited in Flyginnsjøen during a bifurcation event be mixed with material entrained elsewhere in the Glomma catchment?

I commend the authors' application of a rigorous peak detection algorithm and critical assessment of the fidelity with which the sediment record matches the gauged (post-1953) flood record. However, I found Figure 12 and the associated text a bit difficult to follow. For example, Page 18, Lines 12-15 the authors highlight that, overall, there is a good match but also segments that do not correspond, but this analysis would be strengthened had if it was easier to figure out which XRF peak linked to which flood volume bar. One idea would be to colour the vertical flood volume lines and the circle/cross shapes one of two colours when you are confident in their stratigraphic correspondence and the other colour when a match is more difficult to establish. A minor point but, personally, I think the dashed vertical lines denoting each year add substantial clutter to the graph and could be removed. Overall, Figure 12 makes a really important contribution to the paper but it's complicated and uses many different colours and shapes. Improving its aesthetics would really strengthen the paper.

**Comment #4: Summer temperatures as the primary driver**

I would like to see a bit more detail on the physical flood generating mechanisms. From the discussion across pages 23 and 24, it follows that warmer winter air temperatures reduce annual snow storage and, in turn, the magnitude of the spring melt. But I note July temperature is used as the primary meteorological proxy (Figures 17 and 18 and associated text) and I can't figure out why summer

temperature is more important than winter or spring temperature. I would have thought winter temperature would dictate snow volumes and spring temperature would influence rate of melting. I may well have misunderstood but, given this is a fundamental aspect of the interpretation, I suggest modifying the text – and reporting a comparison with winter and/or spring temperatures, if appropriate reconstructions are available - to ensure the reader can follow the process linkages at play. Similarly, Page 23 Line 28 also mentions the importance of winter precipitation. Does a regional precipitation reconstruction exist? I'm not doubting the authors' interpretation but I found myself wondering about the role of other drivers while reading this section so there is scope to tighten up the narrative and analysis here, in my opinion.

**Comments on figures and tables:**

**Figure 3:** do the authors have a photo or aerial/satellite imagery of the bifurcation inflow zone? I am intrigued but I am struggling to visualise what this looks like on the ground. Is there a dry channel or other morphological evidence to indicate this reverse flow occurs on occasion?

**Figure 4 and 5:** Figure 4 suggests the 'normal-flow' inlet to Flyginnsjøen is in the same corner but not in exactly the same place as the inflow under flood conditions, whereas Figure 5 lists only one inlet. I recognise this will have minimal effect but worth clarifying this morphology.

**The sequence of maps are a bit difficult to follow.** For example, it's unclear whether Kongsvinger is a town. Perhaps the inset map in Figure 3 that has one red dot and the boundary of Norway could instead zoom in on a slightly larger area such that Elverum is also visible? Similarly, the locations of the lakes, gauging stations, flood stones, towns and other features mentioned in the text are spread across 3, 4, 5 and 7 and I found myself having to flick back and forth between them.

**Figure 6:** Given its use as a thresheld, it would be worth labelling the 1967 flood in Figure 6

**Figures 14 and 15:** I suggest the authors provide much more context and technical detail in the captions for Figure 14 and 15. I recognise these procedures are explored in the main text but, given their importance to the overall narrative, I think it would be really useful to present ample detail such that both figures can be interpreted on a standalone basis.

**Table 3:** It took me a while to figure out the source of the pairs of correlation coefficients and why some were missing. I understand why the authors have presented the data in this way and it is probably fine to do so with some additional explanation. This might be resolved by writing in the table caption which parameters were measured on which core and also being more explicit on which way round the numbers are reported. Or maybe report the coefficients for one of the cores entirely in italics?

**Minor comments**

**Section 3.3.1:** given the broad audience of the journal, I question whether all readers will be aware of the reasoning behind using and integrating two cores (and indeed two types of corer).

**Page 11, line 27:** the authors state they used a 9-month window in the peak detection algorithm. I like this approach as a way of considering event sequencing but what is the hydrometerological basis for the 9-month window?

**Page 13, line 33:** I suggest the authors report a range of layer thicknesses rather than stating "mm scale".

**Page 19, Line 11:** Judging by eye, there is a more prominent step in flood occurrence rate at 700 yr BP rather than 600 yr BP?

**Page 19, Lines 13-14:** I find the assertion that "high flood frequency in the 18th century is also recorded in the historical flood data (Fig. 6)" to be unconvincing. There are very few data points prior to the 18th century (one?). As long as the authors can be confident the 15th and 18th-century peaks are not triggered by anthropogenic landscape modification, then the sediment record speaks for itself.

**Page 20, Lines 18-20:** I found it difficult to follow the sequence of different approaches applied in Section 4.3. In particular, which is "case ii above" (Line 18) and which is "case iii" (Line 19)?

**Technical corrections:** The manuscript is, for the most part, written in clear, concise prose with ample technical detail. There are a handful of minor typographic errors - mostly inconsistent verb tenses in individual sentences. The clarity of the prose is not affected but no harm in tidying these.

---

## Author Comment (AC1) · 28 Aug 2020

**Response to the comments on the manuscript (HESSD-2019-415) " New flood frequency estimates for the largest river in Norway based on the combination of short and long time series" by Kolbjørn Engeland, Anna Aano, Ida Steffensen, Eivind Støren, Øyvind Paasche**

**This is the authors' answer to the interactive comment posted by Anonymous Referee #1.**

We are very grateful for the excellent review afforded by the anonymous reviewer which we believe will improve our paper.

*Length of the paleoflood record.*
The value of 10 300 comes from Table 5 where the calibrated age based on 14C-dating of the lowest sample is 10259 – 10403 BP (68 % C.I.). We therefore think that 10 300 is a reasonable and robust assessment of the age of the lowest part of the sediment core based on the data used in this study. The paper by Høgås & Longva (2016) does not contradict our age assessment but suggest that the lowest part is not older than 10 400 years. Do note that the age estimate provided by Høgås & Longva (2016) comes with considerable uncertainty as well.

*Section 1*
We agree that a main reason for the increasing of flood damages is the corresponding increase of economic values within the flood plains. We will add this important clarification in the first paragraph of section 1.

*Section 2*
In the figure on the left hand side of Fig. 2, we follow the suggestion by the reviewer and add dashed lines. We will also use different colors in order to increase the contrast between the two. This modification is shown below.

[Figure]

*Specification of the Holocene period:*
The beginning of the Holocene period is in most literature referred by 11 700 years B.P. See e.g Walker et al (2009).

**Section 3 and 4.3**

*Data and periods used for flood frequency analysis:*

The different periods of data used in the flood frequency analysis are as follows

- Systematic data used for flood frequency analysis is 1872 – 1936
- Historical flood data is 1653 – 1871, a length of 219 years
- Paleo-period lasted from 1300 – 1652, i.e. a length of 352 years with 110 flood events in total when we combined paleo- historical and systematic data.
- The paleo record lasted from 1300 – 1871, a period of 571 years with 208 floods when we combined paleo and systematic data.

The period with systematic streamflow data started in 1872. We used data for the period 1872 - 1936 to avoid effects of river regulations on the flood peaks. We did not use the floods from 1966, 1967 and 1995 in the flood frequency analysis. These floods are, however, listed in Table 1 since they are documented on the flood stone. It could be an option, but would require a detailed hydraulic model, to recreate the catchment dynamics with no river regulations. This is beyond the scope for our study. The mean annual flood for the period 1872 – 1936 is 1700 m³/s. For the period 1937-2019 it is 1362 m³/s. A Wilcoxon test indicates that this difference in mean value is significant with a p-value of $2*10^{-8}$ for the zero-hypothesis (i.e. no difference in mean values between the two periods).

The oldest historical flood was observed in 1675. The average waiting time between the historical floods are 22 years. The start of the historical period was therefore set to 1675-22 = 1653 as recommended by Prosdocimi (2018) and Engeland et al. (2018).

We decided to let the paleo-periods start in year 1300, since we have a quasi-stationary period from 1300 until today. Before 1300 the flood frequency is small. We will slightly change the length of the paleo-periods so that they both start in year 1300 and ends the year before the systematic record starts or the start of the historical period.

We will clarify this in both the manuscript and in the figure legends for Figs. 14 and 15.

*Plotting positions:*
The plotting positions in Figure 14 and 15 are based on Hirsch and Stedinger (1987) that is based on the Cunnane plotting position (Cunnane 1978) the exceedance probability $p_i$ of $x_i$

with rank $i$ from a data set with $t$ historical floods representing the historic period $h$, and $s$ systematic floods with $e$ extraordinary floods is given as:

$$p_i = \frac{i - 0.4}{l + 0.2} \cdot \frac{l}{n} \qquad\qquad i = 1,..,l$$

$$p_i = \frac{l}{n} + \frac{n - l}{n} \cdot \frac{i - l - 0.4}{s - e + 0.2} \quad i = l + 1, ..., t + s$$

where $i$ is the rank, $l$ is the number of extraordinary floods ($l = t + e$) and $n$ is the length of the period for which we have information about floods (note that $n = h + s$)

Since we used the systematic streamflow data from 1872-1936, we have 2 systematic floods exceeding the threshold (1916 and 1934) (i.e. $e = 2$) and 9 historical floods (i.e. $t = 9$ and $l = 11$) for the period 1653 - 1871. The length of the systematic record $s = 65$. The length of the historical period $h = 219$ years, thus $n = 284$ years. We did not include the floods from 1966, 1967 and 1995.

The plotting position for the highest flood is 292 years and agrees well with the Figs. 14 and 15.

We will add information about how the plotting positions were calculated.

*Stability of river profile at Elverum:*
The stability of the river profile at Elverum is an important assumption when we assess the discharges of the historical floods. The gauging station was moved around 660 meters in 1969. For the period 1872 – 1968 only one rating curve is used. For the period 1969-2020, the rating curve changed following the flood in 1995. In our study, it was assumed that the river profile at Elverum was stable for the historical period and that the flood in 1789 did not make any substantial changes. This seems to be a reasonable assumption since four large floods occurred during the period 1872 - 1968. We will ad a small discussion on this topic to the revised paper.

**Section 5.2**
The two sub-periods with increased flood peaks during the LIA are indeed interesting, and we thank the reviewer for pointing this out. During this period the average summer temperature for the northern hemisphere did not change substantially. To interpret the dip in flood frequency around year 400 BP remains challenging. Firstly, the temperature anomaly represents an average for the northern hemisphere and not for south-eastern Norway in particular. Secondly, the combination of winter temperature and precipitation might be even more useful for interpreting the flood frequency. Such proxy information is not yet available and therefore limits the possibilities to fully what we observe.
In the revised paper we will add more discussion on flood generating processes and how the available proxy-data can assist us in interpretation of the non-stationarity observed in the paleo-record. See also our answers to Reviewer #2.

**Section 5.3**
We agree that this part of the text explaining how the period for the paleo-data was chosen is a little confusing and we will make an effort to improve the clarity as explained in an earlier comment.

**Small Typos**
All typos will be corrected accordingly. Thanks.

**References**

Cunnane, C. Unbiased plotting positions – a review – comments. Journal of Hydrology 37, 205–222. doi:10.1016/ 0022-1694(78)90017-3, 1978

Engeland, K., Wilson, D., Borsányi, P., Roald, L. and Holmqvist, E. Use of historical data in flood frequency analysis: a case study for four catchments in Norway. Hydrology Research, 49 (2): 466–486. doi: 10.2166/nh.2017.069, 2018

Hirsch, R. M. & Stedinger, J. R. Plotting positions for historical floods and their precision. Water Resources Research 23 (4), 715–727. doi:10.1029/WR023i004p00715, 1987

Høgaas, F. and Longva, O.: Mega deposits and erosive features related to the glacial lake Nedre Glomsjø outburst flood, southeastern Norway, Quaternary Science Reviews, 151, 273–291, doi:10.1016/j.quascirev.2016.09.015, 2016.

Prosdocimi, I. German tanks and historical records: the estimation of the time coverage of ungauged extreme events. Stochastic Environmental Research and Risk Assessment 1–16. doi:10.1007/s00477-017-1418-8, 2018

Walker, M., Johnsen, S., Rasmussen, S. O., Popp, T., Steffensen, J.-P., Gibbard, P., Hoek, W., Lowe, J., Andrews, J., Björck, S., Cwynar, L. C., Hughen, K., Kershaw, P., Kromer, B., Litt, T., Lowe, D. J., Nakagawa, T., Newnham, R., and Schwander, J. 2009. Formal definition and dating of the GSSP (Global Stratotype Section and Point) for the base of the Holocene using the Greenland NGRIP ice core, and selected auxiliary records. *J. Quaternary Sci* ., **Vol. 24** pp. 3–17. ISSN 0267-8179.

---

## Author Comment (AC2) · 28 Aug 2020

Response to the comments on the manuscript (HESSD-2019-415) " New flood frequency estimates for the largest river in Norway based on the combination of short and long time series" by Kolbjørn Engeland, Anna Aano, Ida Steffensen, Eivind Støren, Øyvind Paasche

**This is the authors' answer to the interactive comment posted by Daniel Schillereff.**

We are very grateful for the excellent review afforded by Daniel Schillereff which we believe will improve our paper.

*Comment #1: Structure and content of the introduction*

The suggestions for modifying the introduction are very useful and we will include changes in the revised manuscript.

We will, as suggested, provide more details on how paleo-data can help to reduce uncertainty associated with flood prediction (more data leads to smaller estimation uncertainty) and provide additional insight to flood variability on longer time scales, which can further advance our understanding of how climate change influence flood variability on multiple time scales.

We will also add more information about the time scales involved. Although we do not aim to give a review of different archives suitable for paleoflood reconstructions (e.g. Wilhelm et al., 2018), we will nevertheless in include info on the length of historical records (back to the 16th century) and lake sediment archives (Holocene) available in Norway.

*Comment #2: Human modifications to the landscape and flood stationarity*

Human modifications of the landscape are also important factors for future flood risk and we will add this information to the introduction.

The issue of human influence on the landscape is in this case two-fold and may or may not have sedimentological as well as hydrological influences on the system in question. Changes in land-use and deforestation can impact sediment availability in the catchment, but it can also change the buffering capacity and hence the run-off regime. We believe that the land-use changes caused by the removal of woodland cover that started in the 16th century may have influenced the local erosion and sediment transport of the upstream Glomma catchment (though, note that no data presently exists here), but because this is area represents only a fraction of the total catchment area we think that these 'excess sediments' would be diluted downstream. Another relevant point here is that the downstream gradient of the river Glomma is not steep so sediments can easily be deposited long before they reach the bifurcation point at Kongsvinger. A final point here is that the sediment source for the flood layers deposited in Flyginnsjøen is suggested to be mainly local, and the area around the lake and the location for the bifurcation events itself was not subject to removal of woodland for mining which would reduce the potential influence of anthropogenic influence on the sedimentary budget.

Having said that, we do not rule out the possibility that the removing of woodland since the 16th century amplified the size (and frequency) of floods since forests, in most cases, reduces flood peaks. This, however, require a more regional and systematic vegetation change than that related to mining in the upper Glomma catchment to affect the 154450 km² large catchment.  The 2500-4000 yr BP floodrich period coincidence largely to the bronze age (2500-3700 BP) when settlements and farming expanded in Norway. We see, however, similar flood rich period in other lake sediment records that certainly not are influenced by farming (shown in Fig. 18) and argue that the effect of land-use cannot explain the observed changes in flood frequency. Se also reply to comment #4 on this matter.

**Comment #3: Evaluating the geochemical flood proxy**

We agree that a more thorough sedimentary analysis of potential sediment sources in the catchment could add valuable information to the composition of the recorded flood deposits, and perhaps even denote source areas and thus also flood triggering mechanisms (see eg. Støren et al 2016). Given the size of the Glomma catchment, and the mixing process involved such an approach will require not only sediment samples from the are between the bifurcation threshold and the lake, but also representative samples from potential sedimentary sources in the 154450 km2 catchment for comparison. This would be an interesting exercise, but the workload would be enormous and is well beyond the scope of this paper.

Moreover, during bifurcation events the discharge in the Flyginnsjøen catchment increase with an order of magnitude, dramatically changing the erosion in the area. This change is deemed to mask subtle variability in sediment transport for the Glomma catchment.

Linking the sedimentary signal to specific individual flood events is, as the author recognizes, challenging, and its equally challenging to visualize this link in figure 12. We agree that the dashed vertical lines are not serving any good purpose and have removed these. The suggestion made by the reviewer to add color coding on peaks where we "are confident in their stratigraphic correspondence" is more problematic. Firstly, the plot is already busy, and more colors will add more clutter and probably not improve readability. Secondly, considering the uncertainties in the age-depth model we have not (yet) found a good way to estimate probabilities for correspondence either. Consequently, we rest on the assumption that our 23 recognized peaks in the sedimentary signal corresponds to the 24 bifurcation events over the same period. We argue that this, and the visual correspondence between peaks in K and Ti and peaks in discharge shown in fig 12 are sufficient show causality between bifurcation events and the observed sedimentary signal.

The discussion on uncertainties in the age-depth model and the correspondence between the historical flood events and the sedimentary signal have been elaborated on and we will all the same attempt to improve the aesthetics of figure 12.

**Comment # 4**

We agree with the summary of flood generating mechanisms given by the reviewer. What we see from the flood events in the current climate is

- The flood seasonality in Fig 1 shows that the main flood season lasts from May to June which is locally known as the snow melt season. There are a few floods in the autumn season when periods of sustained rain are the most important flood generating process.

- Investigations of recent large flood events suggest that the key process that generate the largest floods are:

- o Large amounts of snow over large areas available for snow melt. This require high winter precipitation and preferentially low winter temperatures. Several studies indicate a link between snow accumulation and snow melt flood peak/volume in Scandinavia (e.g. Olsson et al, 2018)

- o A cold spring followed by a sudden increase in temperature typically result in high melt rates. Note that the variability and sequence of temperature is important.

- o Large amounts of widespread precipitation combined with snow melt. This factor can be related to spring / summer precipitation.

- o The largest flood in 1789 is not typical for these conditions since it happened in late July in 1789 and intense rain over several days provided the main bulk of the flood water. Snow melt from high altitude areas certainly also contributed, but most likely not to the same extent as for instance the large flood that occurred in 1995 (referred to as 'Vesle-Ofsen')

- Our challenge is to relate these conditions to climatic variables. We speculate that the following climate conditions can enhance floods and boost the flood frequency:

  - o high winter precipitation (more snow will be accumulated throughout the season and also build up perennial snowfields in high mountain areas)

  - o high summer precipitation (especially when it occurs early in the season as such combines with melting of snow)

  - o cold winter and spring temperatures (by pushing back the snow melting well into the summer season the probability of a sudden, substantial raise in temperatures can enhance the potential melting considerably, typically going from under 10 ℃ to over 20 ℃.)

Climate change impacts on floods in Norway is examined in detail by Lawrence (2020), and for the catchment studied here, a decrease in flood magnitude is to be expected. This is in agreement with what paleodata suggests (Støren & Paasche, 2014).

Within Norway there are, however, a variable response of flood magnitudes in snow-dominated catchments and Lawrence (2020) suggests that this reflects the competing effects of increasing winter precipitation and temperature. This anticipated change can lead to either an increase or a decrease in winter snow storage, and/or to an increase in rain-on-snow events throughout the winter half year, depending on the latitude and the elevation of the catchment.

An increase in temperature will also lead to a shift in flood seasonality and flood generating processes (Vormoor et al, 2015, 2016). For Elverum we might expect more frequent autumn floods and less frequent and smaller spring floods.

When it comes to discussing the flood generating mechanisms during the Holocene, we have chosen to compare our results (in fig. 18) with mean reconstructed mean July temperature (Velle et al 2010), and a high-resolution glacier reconstruction (Kvisvik et al 2015) from the mountain areas in the upper Glomma catchment as well as a flood index (Støren et al., 2012) denoting the relative distribution of Holocene floods in Southern Norway. Indeed, comparison to winter, spring and autumn temperature and precipitation reconstruction would have been preferable, but as the reviewer also recognize, there is presently a lack of appropriate climate reconstructions available. We use the plotted paleoreconstructions to discuss the relative influence of changes in summer temperature, but also winter precipitation. We also utilize the flood index (Støren et al 2012) to briefly discuss the effect of changing atmospheric circulation distributing the winter precipitation e.g. causing a likely decrease in winter precipitation c. 2000-1000 cal. yr BP possibly explain the absence of floods recorded for this time interval (page 24 and 25).

We argue, however that summer temperature is highly correlated to winter temperature that can be considered as the main driver of Holocene flood frequencies in the Glomma catchment based on the observed variability both on instrumental and paleo-timescale. The explanation is likely to be linked to lower winter temperatures causing a higher potential for snow melt floods. In such a large catchment, with abundant perennial snow field in the mountains, and the potential for buildup of snowpack over several years, any changes in the flood frequency is deemed to be of regional nature. Consequently, it seems plausible that the effect of raising the 0-isotherm to a higher altitude, the effect of a warmer climate, will significantly change the potential storage of snow and thus flood frequency (see also Støren & Paasche, 2014).

We will to clarify this in the discussion (esp. 5.2)

**Figure 3:**

We do have good aerial photos of the bifurcation inflow zone. A photo will be included in fig. 3.

From the photos one can see the following:

- At Vingersjøen, unambiguous morphological evidence reveal that reverse flow occurs. Especially, since the river that flows out of the lake has actually an inflow delta. The size of this delta has also increased in size.

- For the flow path from Tavern to Flyginnsjøen, there is little geomorphological evidence. The area with intermittent flows is forested or grassland, and the forest cover has increased somewhat the recent years. The most clear evidence are road bridges that has been built across these occasional streams. Evidence of flooding from a bifurcation event is mainly seen as debris in the forested areas.

Figure 4 ad 5: The two inlets into Flyginnsjøen enter the lake at the northern shore and are separated by approximately 10 meters. Note that there is not a delta where Vrangselva enters Flyginnsjøen, indicating that the background sediment influx is low.

**The sequence of maps is a bit difficult to follow.**

Kongsvinger is a town! At least after Norwegian standards. We agree that the sequence of maps is a little difficult to follow. A major challenge is that the Elverum-site for streamflow data and the paleodata at Kongsvinger are located 80 km apart, which is why we think it useful to provide an overview map in Figure 1, detailed maps for the paleo-data in Figures 3 and 4, and detailed map for the historical data in Figure 7. We will give an effort to present details and names in the correct order to simplify the reading of the manuscript, and in Figure 3 we will follow the suggestion to make an zoom in on a smaller area such that Elverum is also visible in the inset map in the upper right corner.

**Figure 6:**

We will label the 1967 flood in Figure 6

*Figures 14 and 15:*
We will present ample detail he figure captions in Figs. 14 and 15. such that both figures can be interpreted on a standalone basis.

**Table                                                                                                      3:**
We will clarify the presentation of Table 3 by adding additional information to the caption and use typographic effects to highlight which correlations are from which core.

**Minor comments**

**Section 3.3.1:**
We will add a few sentences that explain the reasoning for using and integrating two cores.

**Page 11, line 27:**
The hydrometeorological basis for choosing a 9-month window is that in this catchment there is, on average, one major flood event per year. This typically occurs in May/June. Only rarely do we observe large flood events during autumn. Consequently, we expect to detect only one major flood event per year. For locations with more frequent floods, a smaller time window could be more appropriate.

**Page 13, line 33:**
The term "mm scale" is meant as a descriptive term rather than a measure of precise thickness. Given the peak detection algorithm-approach used to recognize flood deposits in the sediment stratigraphy, we have not measured the thickness of all flood layers. This is not a trivial task, since the start and stop of a deposit can be gradual and the signal to noise ratio increase when values are low. We prefer to keep the term "mm-scale" and avoid defining start and stop of individual flood deposits.

**Page 19, Line 11:**
Thanks for pointing this out. We agree that there is a more prominent step in flood occurrence rate at 700 yr BP rather than 600 yr BP, and will change the text accordingly.

**Page 19, Lines 13-14:**
We agree that the comment *"high flood frequency in the 18th century is also recorded in the historical flood data (Fig. 6)"* is speculative and we will delete it.

**Page 20, Lines 18-20:**
We will clarify the different approaches applied in section 4.3 when the paleoflood information is included in the flood frequency analysis.

**References**

Kvisvik, B. C., Paasche, Ø. and Dahl, S. O.: Holocene cirque glacier activity in Rondane, southern Norway, Geomorphology, 11 246, 433–444, doi:10.1016/j.geomorph.2015.06.046, 2015.

Lawrence, D.: Uncertainty introduced by flood frequency analysis in projections for changes in flood magnitudes under a future climate in Norway, Journal of Hydrology: Regional Studies, 28, doi: 10.1016/j.ejrh.2020.100675, 2020

Olsson, J., Uvo, C.B., Foster, K., Yang, W. Technical Note: Initial assessment of a multi-method approach to spring-flood forecasting in Sweden. Hydrol.Earth Syst.Sci 20, 1-9, 2016

Støren, E. N., Kolstad, E. W. and Paasche, Ø.: Linking past flood frequencies in Norway to regional atmospheric circulation anomalies, Journal of Quaternary Science, 27(1), 71–80, doi:10.1002/jqs.1520, 2012.

Støren, E. N. and Paasche, Ø.: Scandinavian floods: From past observations to future trends, Global and Planetary Change,36 113, 34–43, doi:10.1016/j.gloplacha.2013.12.002, 2014.

Velle, G., Bjune, A. E., Larsen, J. and Birks, H. J. B.: Holocene climate and environmental history of Brurskardstjørni, a lake in the catchment of Øvre Heimdalsvatn, south-central Norway, Hydrobiologia, 642(1), 13–34, doi:10.1007/s10750-010-0153-7, 2010

Vormoor, K., Lawrence, D., Heistermann, M., Bronstert, A., Climate change impacts on the seasonality and generation processes of floods – projections and uncertainties for catchments with mixed snowmelt/rainfall regimes. Hydrol. Earth Syst. Sci. 19, 913–931. 2015

Vormoor, K., Lawrence, D., Schlichting, L., Wilson, D., Wong, W.K., Evidence for changes in the magnitude and frequency of observed rainfall vs. Snowmelt driven floods in Norway. J. Hydrol. 538, 33–48, 2016

Wilhelm, B., Ballesteros Canovas, J. A., Corella Aznar, J. P., Kämpf, L., Swierczynski, T., Stoffel, M., Støren, E. and Toonen,W.: Recent advances in paleoflood hydrology: From new archives to data compilation and analysis, Water Security, 1–8, doi: 10.1016/j.wasec.2018.07.001, 2018.

---

## Author Response (AR1)

**Reply to the comments from anonymous reviewer and Daniel Schillereff. on the manuscript (HESSD-2019-415) " New flood frequency estimates for the largest river in Norway based on the combination of short and long time series" by Kolbjørn Engeland, Anna Aano, Ida Steffensen, Eivind Støren, Øyvind Paasche**

We are very grateful for the excellent reviews afforded by the from anonymous reviewer Daniel Schillereff which we believe will improve our paper. Below you find a point to point replay to all comments and the changes in the manuscript are indicated.

Please note that we refer to page and line numbers in the manuscript with tracked changes.

**Reply to the comments from the anonymous reviewer**

**Comment from the referee**
**Length of the paleoflood record:** It is not clear how the authors state the value of 10 300 years from the sediment core. Is it the first dating of a sediment? If yes, according to the dating uncertainty (see a discussion in Dezileau et al., 2014, Geomorphology, 10.1016/j.geomorph.2014.03.017), it may be better to round it to 10 000 years. Another option is to refer to the huge glacial lake outburst flood (reported in section 5.1). It may be the beginning of the paleoflood record. Hogaas and Longva (2016) postulated a date of 10 000 – 10 400 years. So again, a rounded value of 10 000 years may be preferable.

**Author's response**
The value of 10 300 comes from Table 5 where the calibrated age based on 14C-dating of the lowest sample is 10259 – 10403 BP (68 % C.I.). We therefore think that 10 300 is a reasonable and robust assessment of the age of the lowest part of the sediment core based on the data used in this study. The paper by Høgås & Longva (2016) does not contradict our age assessment but suggest that the lowest part is not older than 10 400 years. Note that the age estimate provided by Høgås & Longva (2016) comes with considerable uncertainty as well.

**Author's changes in manuscript**
We keep our age assessment of 10 300 years. To support this age assessment, we have added reference to Table 5 and Figure 11 in the first paragraph of section 4.1 (Page 14, Line 24-25)

**Comment from the referee**
**Section 1:** Page 2, line 3: the main reason for the increasing of flood damages is the increase of economic values within the flood plain. Impact of climate change is of concern, but do not forget that flood risk has two components: flood hazard and flood exposure.

**Author's response**
We agree that a main reason for the increasing of flood damages is the corresponding increase of economic values within the flood plains.

**Author's changes in manuscript**
We have added this important clarification in the first sentences of section 1. (Page 2, line 3-4)

**Comment from the referee**
Figure 2 left: put one curve with dash lines (easier to follow)

**Author's response**
In the figure on the left hand side of Figure 2, we follow the suggestion by the reviewer and add dashed lines. We will also use different colors in order to increase the contrast between the two.

**Author's changes in manuscript**
We have changed Figure 2 (Page 5)

**Comment from the referee**
Page 5, line 5: is the beginning of the Holocene period in Norway exactly 11 700 years ? Give a reference.

**Author's response**
The beginning of the Holocene period is in most literature referred by 11 700 years B.P. See e.g Walker et al (2009).

**Author's changes in manuscript**
We have added the reference to Walker et al (2009) in the first sentence of section 2.2. (Page 5, Line 17)

**Comment from the referee**

**Section 3**
Page 3, lines 8-11: it is not clear within section 3.1 which systematic period is considered. It is written 1871-1937, but in fact, floods larger than x0 = 2533 $m^3$/s have been used (table 1: years 1966, 1967, 1995). If the flood regime was affected by river regulation after 1937, we expect to introduce a correction on the three largest floods after 1937. If not, you can consider the whole period 1871-2019. Please provide the p-value of a statistical test (e.g. Mann Kendall test) with the two periods: 1871-1937 and 1938-2018.

**Author's response**
The period with systematic streamflow data started in 1872. We used data for the period 1872 - 1936 to avoid effects of river regulations on the flood peaks. We did not use the floods from 1966, 1967 and 1995 in the flood frequency analysis. These floods are, however, listed in Table 1 since they are documented on the flood stone. It could be an option, but would require a detailed hydraulic model, to recreate the catchment dynamics with no river regulations. This is beyond the scope for our study. The mean annual flood for the period 1872 – 1936 is 1700 m$^3$/s. For the period 1937-2019 it is 1362 m$^3$/s. A Wilcoxon test indicates that this difference in mean value is significant with a p-value of $2*10^{-8}$ for the zero-hypothesis (i.e. no difference in mean values between the two periods).

**Author's changes in manuscript**
In section 3.1 we have specified that we used only systematic flood observations for the years 1872-1936 (Page 8, Line 3) and added the results of the Wilcoxon test (Page 8, Line 5-6).

**Comment from the referee**

Page 10, lines 10-21: Stability of the geometry of the Glomma river is discussed in section 5.1. May be here in section 3.2.1 or in section 5.1, you could add information on the gaugings. Is the rating curve stable or do we need to use a set of different rating curves according to the gaugings?

**Author's response**
The stability of the river profile at Elverum is an important assumption when we assess the discharges of the historical floods. The gauging station was moved around 660 meters in 1969. For the period 1872 – 1968 only one rating curve is used. For the period 1969-2020, the rating curve changed following the flood in 1995. In our study, it was assumed that the river profile at Elverum was stable for the historical period and that the flood in 1789 did not make any substantial changes. This seems to be a reasonable assumption since four large floods occurred during the period 1872 - 1968. We will ad a small discussion on this topic to the revised paper.
The stability of the river profile at Kongsvinger where the bifurcation event takes place is also an important assumption that is discussed.

**Author's changes in manuscript**
We chose to add this information in the discussion in section 5.1. (Page 23, Line 6-10)

**Comment from the referee**
**Section 4.3:** It is not easy to follow the different options of computation for flood frequency analysis:

- Systematic period: is it 1871-1937 (section 3.1) or 1872-1936 (section 5.3)?
- Historical period: 1653-1872. It may end in 1870 or 1871
- Paleoflood period: 1320-1850. Explain the starting date 1320. If paleoflood and historical floods are in agreement during the overlapping period, we expect to have 1320-1652

**Author's response**
The different periods of data used in the flood frequency analysis are as follows

- Systematic data used for flood frequency analysis is 1872 – 1936
- Historical flood data is 1653 – 1871, a length of 219 years
- Paleo-period lasted from 1300 – 1652, i.e. a length of 352 years with 110 flood events in total when we combined paleo- historical and systematic data.

- The paleo record lasted from 1300 – 1871, a period of 571 years with 208 floods when we combined paleo and systematic data.

The oldest historical flood was observed in 1675. The average waiting time between the historical floods are 22 years. The start of the historical period was therefore set to 1675-22 = 1653 as recommended by Prosdocimi (2018) and Engeland et al. (2018).

We decided to let the paleo-periods start in year 1300, since we have a quasi-stationary period from 1300 until today. Before 1300 the flood frequency is small. We will slightly change the length of the paleo-periods so that they both start in year 1300 and ends the year before the systematic record starts or the start of the historical period.

**Author's changes in manuscript**
We have modified section 4.3 and added Table 6 that summarize the different data sources and periods used for the flood frequency analysis. (Page 22 and 23)

**Comment from the referee**
Plotting positions for figures 14 and 15 are not explained. According to Hirsch and Stedinger (1987), you should consider ALL the floods larger than the threshold x0 over the historical + systematic period, and then the systematic floods lower than x0. We expect to find a set of 14 peak flows (11 from historical period + 3 from the systematic period), and the remaining annual maximum values lower than x0 for the systematic period.

Plotting position of the 1789 flood should be different, according to option 1 or 2 (largest flood since 1653 or since 10 000 years).

**Author's response**
The plotting positions in Figure 14 and 15 are based on Hirsch and Stedinger (1987) that is based on the Cunnane plotting position (Cunnane 1978) the exceedance probability $p_i$ of $x_i$ with rank $i$ from a data set with $t$ historical floods representing the historic period $h$, and $s$ systematic floods with $e$ extraordinary floods is given as:

$$p_i = \frac{i - 0.4}{l + 0.2} \cdot \frac{l}{n} \qquad i = 1, .., l$$

$$p_i = \frac{l}{n} + \frac{n - l}{n} \cdot \frac{i - l - 0.4}{s - e + 0.2} \quad i = l + 1, ..., t + s$$

where $i$ is the rank, $l$ is the number of extraordinary floods ($l = t + e$) and $n$ is the length of the period for which we have information about floods (note that $n = h + s$)

Since we used the systematic streamflow data from 1872-1936, we have 2 systematic floods exceeding the threshold (1916 and 1934) (i.e. $e = 2$) and 9 historical floods (i.e. $t = 9$ and $l = 11$) for the period 1653 - 1871. The length of the systematic record $s = 65$. The length of the historical period $h = 219$ years, thus $n = 284$ years. We did not include the floods from 1966, 1967 and 1995.

The plotting position for the highest flood is 292 years and agrees well with the Figs. 14 and 15.

**Author's changes in manuscript**
We have added section 3.4.2 that summarize the plotting position. We refer to 3.4.2 in the captions of Figure 14 and 15.
 (Page 13 line 36-41 and Page 14 line 2-3)

**Comment from the referee**
On figure 5, it is not easy to understand which curves relate to cases (i), (ii) and (iii) vs systematic floods included, all floods included, paleoflood included.

**Author's response**
We think this comment refers to Figure 15. We agree that this is unclear and have improved the Figure captions and the text and also added Table 6 that summarize the period for all data sources.

**Author's changes in manuscript**
The caption of Figure 15 is improved as well as the text summarizing the results. (Page 22, line 7-22). We have also added Table 6 that summarize the time period covered for all data sources.

**Comment from the referee**

**Section 5.2**
On figure 17, it is very interesting to see the flood rich period during the LIA with low temperature (and the contrary before 500 cal. yr BP). The authors could add a comment on the fact that we have two peaks during LIA but the temperature did not significantly change.

**Author's response**
The two sub-periods with increased flood peaks during the LIA are indeed interesting, and we thank the reviewer for pointing this out. During this period the average summer temperature for the northern hemisphere did not change substantially. To interpret the dip in flood frequency around year 400 BP remains challenging. Firstly, the temperature anomaly represents an average for the northern hemisphere and not for south-eastern Norway in particular. Secondly, the combination of winter temperature and precipitation might be even more useful for interpreting the flood frequency. Such proxy information is not yet available and therefore limits the possibilities to fully what we observe.
In the revised paper we will add more discussion on flood generating processes and how the available proxy-data can assist us in interpretation of the non-stationarity observed in the paleo-record. See also our answers to Reviewer #2.

**Author's changes in manuscript**

We have modified the discussion in section 5.2 (Page 25, lines 24-26)

**Comment from the referee**

**Section 5.3**
Page 26, line 6: it remains unclear how the 600 year period is chosen. Additionally, according to section 4.3, it a rather a 530 year period (1320-1850) or a 332 year period (1320-1652)

**Author's response**
We agree that this part of the text explaining how the period for the paleo-data was chosen is a little confusing and we will make an effort to improve the clarity as explained in an earlier comment.

**Author's changes in manuscript**
We have changed the 6[th] paragraph in section 5.3 (Page 29, line 27)

**Comment from the referee**
**Small typos**
Page 2, line 18: they require that buildings
Page 2, line 22: up to 1000-year floods… (see Lovdata, 2010, and TEK17, 2018 for regulations
Page 3, lines 24-25: (Hanssen-Bauer et al., 2017
Page 5, line 16: the flow direction reverses
Page 5, line 24: in Fig. 5
Page 6, line 6: (after Hegge, 1968)
Page 10, line 4: reference of Ostmoe (1985) is missing in section 7
Figure 8: Legend "Max discharge"
Page 11, line 2: (Renberg and Hansson, 2008). Samples
Page 11, line 39 add "if k=0 (Gumbel distribution)" on the second part of equation (1)
Page 22, line 18: A second assumption is that the river
Page 22, line 20: Hogaas and Langva (2016) give an age of 10 – 10.4 cal ka BP in conclusion
Page 22, line 22: Based on Klaeboe (1946) and Hegge (1968) the threshold
Page 22, line 26: According to Hegge (1968), the… in 1967 CE
Page 22, line 38: and regional
Page 23, line 15: reference of Moberg et al. (2015) is missing in section 7
Page 24, line 14: Velle et al., 2010
Page 24, line 22: and in the central alps
Page 25, line 4: Storen et al. (2012)
Page 26, line 10: the threshold for historical floods is too low
Page 27, line 25: 7. References

**Author's response**
We have corrected all these typos in the revised manuscript.

**Author's changes in manuscript**
We have corrected all these typos in the revised manuscript

**Reply to the comments from Daniel Schillereff**

**Comment from the referee**
**Comment #1: Structure and content of the introduction** : Given the broad audience of HESS, and the likelihood that some (many?) readers may be more accustomed to studying recent datasets, I suggest the authors incorporate more detail on the palaeodata in the introduction. I specifically suggest the authors elaborate on the ways in which palaeo data can help address the "two reasons" outlined on Page 2, Lines 23-26. Whilst many readers will have a sound knowledge of return periods and design flood estimates, applying palaeo data to these processes – and the value of doing so - may be quite new.
Similarly, I think it would be useful for the authors to say more about the timescales involved: how far back in time can palaeohydrological data be obtained from lake sediments (page 2, lines 42-43) and historical records (page 3, lines 5-11)? This would give readers a platform of knowledge that will be helpful when they engage with the return period calculations later in the paper. Likewise, I think the objectives (Page 3, lines 39-43 and on to Page 4) could be a bit more specific and pay particular attention to the timescale of your analysis
I also query whether Page 3, lines 5-11 could be slimmed down. To keep the focus on your work, perhaps mention briefly that there are a range of historical archives but place particular emphasis on epigraphic sources – especially flood stones - as that is the sort of data used later in this study

**Author's response**
The suggestions for modifying the introduction are very useful and we will include changes in the revised manuscript.

We will, as suggested, provide more details on how paleo-data can help to reduce uncertainty associated with flood prediction (more data leads to smaller estimation uncertainty) and provide additional insight to flood variability on longer time scales, which can further advance our understanding of how climate change influence flood variability on multiple time scales.

We will also add more information about the time scales involved. Although we do not aim to give a review of different archives suitable for paleoflood reconstructions (e.g. Wilhelm et al., 2018), we will nevertheless in include info on the length of historical records (back to the 16[th] century) and lake sediment archives (Holocene) available in Norway.

**Author's changes in manuscript**
We have modified the introduction (Page 3, Line 5 – 18)

**Comment from the referee**
I am pleased the authors mention the possibility that changes in land-use could play a role and, similarly, I was really pleased that the authors explicitly assess whether the fluvial system behaviour would have allowed bifurcation events throughout the Holocene (Page 22, Lines 18-30). I think there is scope to go further in ruling out the possibility that changes in flood frequency are driven by rather than amplified by human activity.
Page 3, line 24: it is important to state here that non-stationarity needn't only be a response to climatic variability. Human alteration of the land surface can also create a non-stationary fluvial system.
Page 5, Lines 2-5 talk about 'noteworthy land-use changes during the last 400 years, and specifically the removal of woodland cover". This is returned to briefly on Page 25, Lines 6-15

but I think this needs a more critical and in-depth evaluation. The authors acknowledge, for example, the notable rise in flood frequency rises around 500 yr BP, which happens to coincide with the assertion on Page 25 Line 8 "The mining industry that started in Norway in the 16th century required a large amount of timber which resulted in widespread deforestation also in Glomma's catchment". How confident are the authors that widespread deforestation amplified the climate driver but was not a driver of sedimentological change in its own right?

Similarly, I think the interpretation of external drivers, especially through the 2500-4000 yr BP flood-rich period, would be strengthened considerably if human interference could be ruled out. In Britain, for example, a number of fluvial and palaeolimnological studies show widespread mobilisation of sediments at that time resulting from settlement expansion. I have no idea about the mid- to late-Holocene history of human occupation in southern Norway but presenting or referring to data ruling this out possibility would really strengthen the case.

**Author's response**

Human modifications of the landscape are also important factors for future flood risk and we will add this information to the introduction.

The issue of human influence on the landscape is in this case two-fold and may or may not have sedimentological as well as hydrological influences on the system in question. Changes in land-use and deforestation can impact sediment availability in the catchment, but it can also change the buffering capacity and hence the run-off regime. We believe that the land-use changes caused by the removal of woodland cover that started in the 16th century may have influenced the local erosion and sediment transport of the upstream Glomma catchment (though, note that no data presently exists here), but  because this is area represents only a fraction of the total catchment area we think that these 'excess sediments' would be diluted downstream. Another relevant point here is that the downstream gradient of the river Glomma is not steep so sediments can easily be deposited long before they reach the bifurcation point at Kongsvinger. A final point here is that the sediment source for the flood layers deposited in Flyginnsjøen is suggested to be mainly local, and the area around the lake and the location for the bifurcation events itself was not subject to removal of woodland for mining which would reduce the potential influence of anthropogenic influence on the sedimentary budget.

Having said that, we do not rule out the possibility that the removing of woodland since the 16th century amplified the size (and frequency) of floods since forests, in most cases, reduces flood peaks. This, however, require a more regional and systematic vegetation change than that related to mining in the upper Glomma catchment to affect the 154450 km$^2$ large catchment.  The 2500-4000 yr BP flood-rich period coincidence largely to the bronze age (2500-3700 BP) when settlements and farming expanded in Norway. We see, however, similar flood rich period in other lake sediment records that certainly not are influenced by farming (shown in Fig. 18) and argue that the effect of land-use cannot explain the observed changes in flood frequency. Se also reply to comment #4 on this matter.

**Author's changes in manuscript**

We have modified section 5.2 in the discussion chapter (Page 27, Line 20 – Page 28, Line 8).

**Comment from the referee**
**Comment #3: Evaluating the geochemical flood proxy**
Overall, I find the proxy reconstruction to be convincing. I do wonder whether it could be strengthened by providing some information on the geochemical composition of the glacio-fluvial material between the two lakes. The authors state the Glomma catchment is the largest in Norway. To what extent will sediments being deposited in Flyginnsjøen during a bifurcation event be mixed with material entrained elsewhere in the Glomma catchment?

I commend the authors' application of a rigorous peak detection algorithm and critical assessment of the fidelity with which the sediment record matches the gauged (post-1953) flood record. However, I found Figure 12 and the associated text a bit difficult to follow. For example, Page 18, Lines 12-15 the authors highlight that, overall, there is a good match but also segments that do not correspond, but this analysis would be strengthened had if it was easier to figure out which XRF peak linked to which flood volume bar. One idea would be to colour the vertical flood volume lines and the circle/cross shapes one of two colours when you are confident in their stratigraphic correspondence and the other colour when a match is more difficult to establish. A minor point but, personally, I think the dashed vertical lines denoting each year add substantial clutter to the graph and could be removed. Overall, Figure 12 makes a really important contribution to the paper but it's complicated and uses many different colours and shapes. Improving its aesthetics would really strengthen the paper.

**Author's response**
We agree that a more thorough sedimentary analysis of potential sediment sources in the catchment could add valuable information to the composition of the recorded flood deposits, and perhaps even denote source areas and thus also flood triggering mechanisms (see eg. Støren et al 2016). Given the size of the Glomma catchment, and the mixing process involved such an approach will require not only sediment samples from the are between the bifurcation threshold and the lake, but also representative samples from potential sedimentary sources in the 154450 km2 catchment for comparison. This would be an interesting exercise, but the workload would be enormous and is well beyond the scope of this paper.

Moreover, during bifurcation events the discharge in the Flyginnsjøen catchment increase with an order of magnitude, dramatically changing the erosion in the area. This change is deemed to mask subtle variability in sediment transport for the Glomma catchment.

Linking the sedimentary signal to specific individual flood events is, as the reviewer recognizes, challenging, and its equally challenging to visualize this link in figure 12. We agree that the dashed vertical lines are not serving any good purpose and have removed these. The suggestion made by the reviewer to add color coding on peaks where we "are confident in their stratigraphic correspondence" is more problematic. Firstly, the plot is already busy, and more colors will add more clutter and probably not improve readability. Secondly, considering the uncertainties in the age-depth model we have not (yet) found a good way to estimate probabilities for correspondence either. Consequently, we rest on the assumption that our 23 recognized peaks in the sedimentary signal corresponds to the 24 bifurcation events over the same period. We argue that this, and the visual correspondence between peaks in K and Ti and peaks in discharge shown in fig 12 are sufficient show causality between bifurcation events and the observed sedimentary signal.

**Author's changes in manuscript**
The discussion on uncertainties in the age-depth model and the correspondence between the historical flood events and the sedimentary signal have been elaborated on in section 5.1 in the discussion (Page 23, line 18-22) and we have improved the aesthetics of figure 12 (Page 19).

**Comment from the referee**
Comment #4: Summer temperatures as the primary driver I would like to see a bit more detail on the physical flood generating mechanisms. From the discussion across pages 23 and 24, it follows that warmer winter air temperatures reduce annual snow storage and, in turn, the magnitude of the spring melt. But I note July temperature is used as the primary meteorological proxy (Figures 17 and 18 and associated text) and I can't figure out why summer temperature is more important than winter or spring temperature. I would have thought winter temperature would dictate snow volumes and spring temperature would influence rate of melting. I may well have misunderstood but, given this is a fundamental aspect of the interpretation, I suggest modifying the text – and reporting a comparison with winter and/or spring temperatures, if appropriate reconstructions are available - to ensure the reader can follow the process linkages at play. Similarly, Page 23 Line 28 also mentions the importance of winter precipitation. Does a regional precipitation reconstruction exist? I'm not doubting the authors' interpretation but I found myself wondering about the role of other drivers while reading this section so there is scope to tighten up the narrative and analysis here, in my opinion.

**Author's response**
We agree with the summary of flood generating mechanisms given by the reviewer. What we see from the flood events in the current climate is

- The flood seasonality in Fig 1 shows that the main flood season lasts from May to June which is locally known as the snow melt season. There are a few floods in the autumn season when periods of sustained rain are the most important flood generating process.

- Investigations of recent large flood events suggest that the key process that generate the largest floods are:

    o Large amounts of snow over large areas available for snow melt. This require high winter precipitation and preferentially low winter temperatures. Several studies indicate a link between snow accumulation and snow melt flood peak/volume in Scandinavia (e.g. Olsson et al, 2018)

    o A cold spring followed by a sudden increase in temperature typically result in high melt rates. Note that the variability and sequence of temperature is important.

    o Large amounts of widespread precipitation combined with snow melt. This factor can be related to spring / summer precipitation.

    o The largest flood in 1789 is not typical for these conditions since it happened in late July in 1789 and intense rain over several days provided the main bulk of the flood water. Snow melt from high altitude areas certainly also contributed, but most likely not to the same extent as for instance the large flood that occurred in 1995 (referred to as 'Vesle-Ofsen')

- Our challenge is to relate these conditions to climatic variables. We speculate that the following climate conditions can enhance floods and boost the flood frequency:
    - high winter precipitation (more snow will be accumulated throughout the season and also build up perennial snowfields in high mountain areas)
    - high summer precipitation (especially when it occurs early in the season as such combines with melting of snow)
    - cold winter and spring temperatures (by pushing back the snow melting well into the summer season the probability of a sudden, substantial raise in temperatures can enhance the potential melting considerably, typically going from under 10 °C to over 20 °C.)

Climate change impacts on floods in Norway is examined in detail by Lawrence (2020), and for the catchment studied here, a decrease in flood magnitude is to be expected. This is in agreement with what paleodata suggests (Støren & Paasche, 2014).

Within Norway there are, however, a variable response of flood magnitudes in snow-dominated catchments and Lawrence (2020) suggests that this reflects the competing effects of increasing winter precipitation and temperature. This anticipated change can lead to either an increase or a decrease in winter snow storage, and/or to an increase in rain-on-snow events throughout the winter half year, depending on the latitude and the elevation of the catchment.

An increase in temperature will also lead to a shift in flood seasonality and flood generating processes (Vormoor et al, 2015, 2016). For Elverum we might expect more frequent autumn floods and less frequent and smaller spring floods.

When it comes to discussing the flood generating mechanisms during the Holocene, we have chosen to compare our results (in fig. 18) with mean reconstructed mean July temperature (Velle et al 2010), and a high-resolution glacier reconstruction (Kvisvik et al 2015) from the mountain areas in the upper Glomma catchment as well as a flood index (Støren et al., 2012) denoting the relative distribution of Holocene floods in Southern Norway. Indeed, comparison to winter, spring and autumn temperature and precipitation reconstruction would have been preferable, but as the reviewer also recognize, there is presently a lack of appropriate climate reconstructions available. We use the plotted paleo-reconstructions to discuss the relative influence of changes in summer temperature, but also winter precipitation. We also utilize the flood index (Støren et al 2012) to briefly discuss the effect of changing atmospheric circulation distributing the winter precipitation e.g. causing a likely decrease in winter precipitation c. 2000-1000 cal. yr BP possibly explain the absence of floods recorded for this time interval (page 24 and 25).

We argue, however that summer temperature is highly correlated to winter temperature that can be considered as the main driver of Holocene flood frequencies in the Glomma catchment based on the observed variability both on instrumental and paleo-timescale. The explanation is likely to be linked to lower winter temperatures causing a higher potential for snow melt floods. In such a large catchment, with abundant perennial snow field in the mountains, and the potential for buildup of snowpack over several years, any changes in the flood frequency is deemed to be of regional nature. Consequently, it seems plausible that the effect of raising the 0-isotherm to a higher altitude, the effect of a warmer climate, will significantly change the potential storage of snow and thus flood frequency (see also Støren & Paasche, 2014).

**Author's changes in manuscript**
We have modified section 5.2 in the discussion chapter. (Pag 24 Line 35 –Page 25 Line 26)

**Comment from the referee**
Figure 3: do the authors have a photo or aerial/satellite imagery of the bifurcation inflow zone? I am intrigued but I am struggling to visualise what this looks like on the ground. Is there a dry channel or other morphological evidence to indicate this reverse flow occurs on occasion?
We do have good aerial photos of the bifurcation inflow zone. A photo will be included in fig. 3.

**Author's response**
From the photos one can see the following:

- At Vingersjøen, unambiguous morphological evidence reveal that reverse flow occurs. Especially, since the river that flows out of the lake has actually an inflow delta. The size of this delta has also increased in size.

- For the flow path from Tavern to Flyginnsjøen, there is little geomorphological evidence. The area with intermittent flows is forested or grassland, and the forest cover has increased somewhat the recent years. The most clear evidence are road bridges that has been built across these occasional streams.

**Author's changes in manuscript.**
We have modified Figure 3 by adding two areal photos of the area close to Kongsvinger (Page 6).

**Comment from the referee**
Figure 4 and 5: Figure 4 suggests the 'normal-flow' inlet to Flyginnsjøen is in the same corner but not in exactly the same place as the inflow under flood conditions, whereas Figure 5 lists only one inlet. I recognise this will have minimal effect but worth clarifying this morphology.

**Author's response**
Figure 4 ad 5: The two inlets into Flyginnsjøen enter the lake at the northern shore and are separated by approximately 30 meters. Note that there is not a delta where Vrangselva enters Flyginnsjøen, indicating that the background sediment influx is low.

**Author's changes in manuscript**
We have added this information in the text and in the captions of Figure 4 and 5. We have modified Figure 5 to show the two inlets into Flyginnsjøen (Page 5 Line 29, Page 6 Line 1 and Page 7).

**Comment from the referee**
The sequence of maps are a bit difficult to follow. For example, it's unclear whether Kongsvinger is a town. Perhaps the inset map in Figure 3 that has one red dot and the boundary of Norway could instead zoom in on a slightly larger area such that Elverum is also visible? Similarly, the locations of the lakes, gauging stations, flood stones, towns and other features mentioned in the text are spread across 3, 4, 5 and 7 and I found myself having to flick back and forth between them.

**Author's response**
Kongsvinger is a town! At least after Norwegian standards. We agree that the sequence of maps is a little difficult to follow. A major challenge is that the Elverum-site for streamflow data and the paleodata at Kongsvinger are located 80 km apart, which is why we think it useful to provide an overview map in Figure 1, detailed maps for the paleo-data in Figures 3 and 4, and detailed map for the historical data in Figure 7. We will give an effort to present details and names in the correct order to simplify the reading of the manuscript, and in Figure 3 we will follow the suggestion to make an zoom in on a smaller area such that Elverum is also visible in the inset map in the upper right corner.

**Author's changes in manuscript.**
We have modified Figure 3 by zoom in on a smaller area so that Elverum is visible in the inset map in the upper right corner. We have also added one name that is used in the text (Vrangselva) , and two areal photos of the area close to Kongsvinger (Page 6). We have also changed Figure 7 and the text and use the name 'Klokkerfossen' for the location of one the flood stones (Page 10).

**Comment from the referee**
**Figure 6:** Given its use as a threshold, it would be worth labelling the 1967 flood in Figure 6

**Author's response** '
The 1967 flood is now labelled in Figure 6

**Author's changes in manuscript.**
We have modified Figure 6 including the threshold, added the discharge for the threshold and the length of the historical period. All details are explained in the figure caption. (Page 8)

**Comment from the referee**
Figures 14 and 15: I suggest the authors provide much more context and technical detail in the captions for Figure 14 and 15. I recognise these procedures are explored in the main text but, given their importance to the overall narrative, I think it would be really useful to present ample detail such that both figures can be interpreted on a standalone basis.

**Author's response** '
We agree

**Author's changes in manuscript.**
We have added more details in the figure captions in  Figs. 14 and 15. such that both figures can be interpreted on a standalone basis. (Page 21 and 22)

**Comment from the referee**
**Table 3:** It took me a while to figure out the source of the pairs of correlation coefficients and why some were missing. I understand why the authors have presented the data in this way and it is probably fine to do so with some additional explanation. This might be resolved by writing in the table caption which parameters were measured on which core and also being more explicit on which way round the numbers are reported. Or maybe report the coefficients for one of the cores entirely in italics?

**Author's response** '
We have clarified the presentation of Table 3 by adding additional information to the caption and use typographic effects to highlight which correlations are from which core.

**Author's changes in manuscript.**
We have change Table 3 accordingly. (Page 16)

**Minor comments**

**Comment from the referee**
**Section 3.3.1:** given the broad audience of the journal, I question whether all readers will be aware of the reasoning behind using and integrating two cores (and indeed two types of corer).

**Author's response**
We agree and have add a few sentences explaining the reasoning for using and integrating two cores.

**Author's changes in manuscript.**
We have modified section 3.3.1 (Page 11, Line 14-17)

**Comment from the referee**
**Page 11, line 27:** the authors state they used a 9-month window in the peak detection algorithm. I like this approach as a way of considering event sequencing but what is the hydrometerological basis for the 9-month window?

**Author's response**
First one clarification: In section 3.3.1 the 9 months refer to the minimum time lag between two succeeding flood events, it is not the size of the time window used in the flood detection algorithm. The hydrometeorological basis for choosing a 9-month time lag is that in this catchment there is, on average, one major flood event per year. This typically occurs in May/June. Only rarely do we observe large flood events during autumn. Consequently, we expect to detect only one major flood event per year. For locations with more frequent floods, a smaller time window could be more appropriate.

**Author's changes in manuscript.**
Se section 3.3.1 page 12 line 11-15 in the revised manuscript

**Comment from the referee**
**Page 13, line 33:** I suggest the authors report a range of layer thicknesses rather than stating "mm scale".

**Author's response**

The term "mm scale" is meant as a descriptive term rather than a measure of precise thickness. Given the peak detection algorithm-approach used to recognize flood deposits in the sediment stratigraphy, we have not measured the thickness of all flood layers. This is not a trivial task, since the start and stop of a deposit can be gradual and the signal to noise ratio increase when values are low. We prefer to keep the term "mm-scale" and avoid defining start and stop of individual flood deposits.

**Author's changes in manuscript.**

None

**Comment from the referee**

**Page 19, Line 11:** Judging by eye, there is a more prominent step in flood occurrence rate at 700 yr BP rather than 600 yr BP?

**Author's response**

Thanks for pointing this out. We agree that there is a more prominent step in flood occurrence rate at 700 yr BP rather than 600 yr BP, and will change the text accordingly.

**Author's changes in manuscript.**

We have changed section 4.2 (Page 20, line 12)

**Comment from the referee**

**Page 19, Lines 13-14:** I find the assertion that "high flood frequency in the 18th century is also recorded in the historical flood data (Fig. 6)" to be unconvincing. There are very few data points prior to the 18th century (one?). As long as the authors can be confident the 15th and 18th-century peaks are not triggered by anthropogenic landscape modification, then the sediment record speaks for itself.

**Author's response**

We agree that the comment *"high flood frequency in the 18th century is also recorded in the historical flood data (Fig. 6)"* is speculative and we will delete it.

**Author's changes in manuscript.**

We have changed section 4.2 (Page 20, lines 14-15)

**Comment from the referee**

**Page 20, Lines 18-20:** I found it difficult to follow the sequence of different approaches applied in Section 4.3. In particular, which is "case ii above" (Line 18) and which is "case iii" (Line 19)?

**Author's response**

We will clarify the different approaches applied in section 4.3 when the paleoflood information is included in the flood frequency analysis.

-

**Author's changes in manuscript.**
We have modified section 4.3 and added table 6 to clarify the different approaches. (Page 21 and 22)

[revised manuscript text omitted]